# Towards Fast Safe Online Reinforcement Learning via Policy Finetuning

**Keru Chen**                                                               *kchen234@asu.edu*
*School of Electrical, Computer and Energy Engineering*
*Arizona State University*

**Honghao Wei**                                                         *honghao.wei@wsu.edu*
*School of Electrical Engineering and Computer Science*
*Washington State University*

**Zhigang Deng**                                                       *zdeng4@central.uh.edu*
*Department of Computer Science*
*University of Houston*

**Sen Lin**[*]                                                           *slin50@central.uh.edu*
*Department of Computer Science*
*University of Houston*

**Reviewed on OpenReview:** *https://openreview.net/forum?id=1SO7vmLFUq*

## Abstract

High costs and risks involved in extensive environmental interactions hinder the practical application of current online safe reinforcement learning (RL) methods. Inspired by recent successes in offline-to-online (O2O) RL, it is crucial to explore whether offline safe RL can be leveraged to facilitate faster and safer online learning, a direction that has yet to be fully investigated. To fill this gap, we first show that naively applying existing O2O algorithms from standard RL would not work well in safe RL due to two unique challenges: *erroneous Q-estimations*, resulted from offline-online objective mismatch and offline cost sparsity, and *Lagrangian mismatch*, resulted from difficulties in aligning Lagrange multipliers between offline and online policies. To address these challenges, we introduce **Marvel**, the first policy-finetuning based framework for O2O safe RL, comprising two key components that work in concert: *Value Pre-Alignment* to align the learned Q-functions with the online objective before finetuning, and *Adaptive PID Control* to effectively adjust the Lagrange multipliers during finetuning. Extensive experiments demonstrate the superior performance of Marvel over related baselines.

## 1 Introduction

Safe reinforcement learning (safe RL) (Gu et al., 2022; Garcıa & Fernández, 2015) prioritizes not only reward maximization but also the adherence to safety constraints, enhancing its applicability in real-world scenarios. For instance, an autonomous vehicle must reach its destination without exceeding a fuel limit. However, solving safe online RL from scratch in fields such as robotics (Brunke et al., 2022; Kiran et al., 2021) and healthcare (Yu et al., 2021; Qayyum et al., 2020) is often prohibitive, due to the high risks and costs of extensive environment interactions. To address this, offline safe RL (Zheng et al., 2024; Ray et al., 2019) has been introduced, enabling learning safe policies from a static dataset without online interactions. Yet, offline

---

[*]Corresponding author.

safe RL faces its own limitations (Ghosh et al., 2022): it typically shows limited performance, relies on the quality of the offline dataset, and suffers from the impact of out-of-distribution (OOD) actions, restricting its effectiveness across varying scenarios.

The pretraining-and-finetuning paradigm is a well-known strategy in the fields of computer vision and natural language processing, for enabling fast online learning based on offline pretrained models. Following a similar line, offline-to-online RL (O2O RL) in the unconstrained setting (Nair et al., 2020; Wang et al., 2024; Zhang et al., 2023b) and imitation learning (Yue et al., 2024; Ross et al., 2011) have recently gained prominence. These approaches utilize policies (including Q-functions) derived from offline learning, along with offline datasets, to expedite online finetuning, which effectively avoids extensive environmental interactions in training policies from scratch. Note that in *offline-to-online safe RL*, the goal is not large-scale pretraining but rather to obtain a safe warm start from offline data and perform limited online finetuning under safety constraints. Thus motivated, a key insight is leveraging the pretraining-and-finetuning paradigm can also potentially facilitate more efficient and practical online safe RL, which however has not been fully explored in the literature. To fill this gap, we seek to answer the following question: *Can we design an effective offline-to-online approach for safe RL to address the limitations of both online and offline safe RL, thereby enabling fast online safe policy learning?*

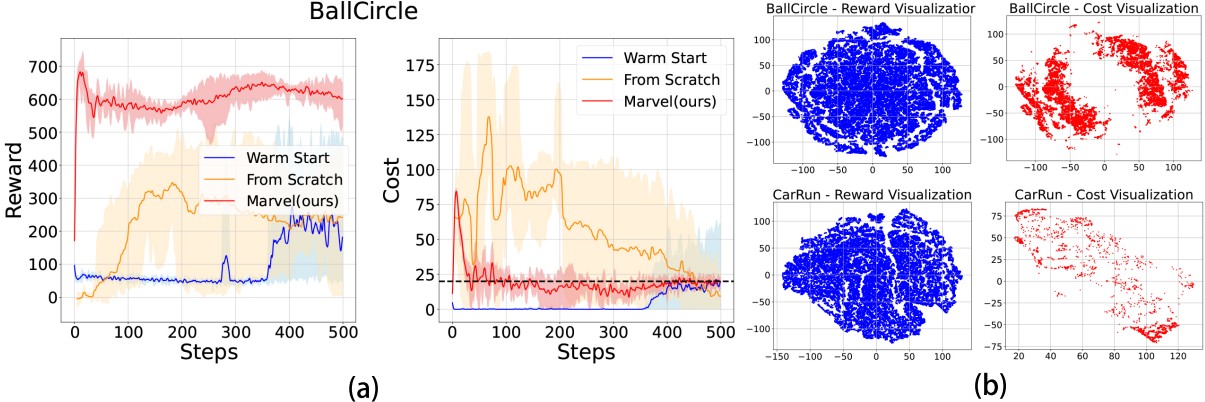

Figure 1: "Steps" on the x-axis represent the number of policy gradient updates (i.e., optimizer updates). For each update, the agent interacts with the environment for 3 episodes. This convention is followed in the subsequent figures. **In (a)**, we evaluate these methods in BallCircle from Bullet Safety Gym (Gronauer, 2022), with a cost limit of 20. As shown, while "Warm Start" begins with a reasonably good initial policy, it performs poorly and overly conservatively, even worse than "From Scratch" where the policy and Q-functions are initialized randomly. This implies that directly finetuning the pretrained policy and Q-functions may hinder online learning. In contrast, "Marvel" achieves impressive results, finding a policy with much higher return in just a few online steps while adhering to the cost limit. **In (b)**, t-SNE visualization of state in the environment, reduced to 2D space. Each point represents a state, with rewards uniformly distributed across the space, while costs are sparse, appearing as isolated points or clusters, reflecting their limited association with states.

However, achieving this is highly nontrivial, and simply applying existing O2O algorithms in conventional RL would not work well here due to unique challenges in safe RL. In Fig. 1 (a), 'Warm Start' refers to using the offline pretrained policy and Q-networks directly to initialize an online safe RL algorithm. 'From Scratch' refers to purely online safe RL training. As illustrated, directly finetuning the offline pretrained policy and Q-functions by using standard online safe RL often results in suboptimal performance and, in some cases, complete training failures.

The reasons behind this phenomenon are as follows: a) *Erroneous Q-estimations resulting from objective mismatch and offline cost sparsity.* To avoid explorations beyond offline data and reduce the extrapolation errors, offline safe RL algorithms typically introduce additional regularizations in the objective function to push up the cost estimates of OOD actions, e.g., VOCE (Guan et al., 2024) and CPQ (Xu et al., 2022),

leading to a different overall objective from standard online safe RL. More critically, the majority of state-actions in offline datasets for safe RL usually are safe with zero cost (Fig. 1 (b)), resulting in a pretrained cost Q-function that predicts extremely low cost for most in-distribution (IND) state-actions. By erroneously giving high values for OOD state-actions and low values for IND state-actions, the pretrained cost Q-function will conservatively force the online finetuning to stay in the state-action space similar to offline dataset and be reluctant to explore (e.g., cost of "Warm Start" in Fig. 1). b) *Mismatch of Lagrange multipliers.* Many online safe RL algorithms (Chow et al., 2018a; Achiam et al., 2017) solve the constrained optimization problem based on the primal-dual approach, which requires a synchronous update of Lagrange multipliers. Nonetheless, initial values for these multipliers that match the offline policies cannot be obtained from offline safe RL precisely. Using traditional dual ascent methods to update the Lagrange multipliers may result in slow online learning even with accurately estimated Q-functions, ultimately degrading the performance of the learned policy. In this work, we seek to design an effective O2O framework for safe RL by addressing these two challenges above.

The main contribution of this work is the **warM-stArt safe Reinforcement learning with Value prEaLignment (Marvel)** framework, consisting of two key components: Value Pre-Alignment (VPA) and Adaptive PID Control (aPID). VPA re-aligns pretrained Q-functions to closely match true Q-values of the offline policy under the online learning objective by re-evaluating the offline policy based on offline data alone. To further compensate VPA and address the multiplier mismatch problem, aPID quickly adapts Lagrange multipliers based on cost violations instead of directly finding the best initial multipliers. Experimental results demonstrate Marvel's ability to quickly learn a safe, high-reward policy with minimal online interactions, marking it as a pioneering solution in O2O safe RL that integrates seamlessly with existing offline and online safe RL methods.

## 2 Preliminaries

**Constrained Markov Decision Process.** We consider a standard constrained Markov Decision Process (CMDP) (Altman, 2021), defined by a tuple $(S, A, T, R, C, \gamma, \eta, c_{th})$. Here $S$ is the state space, $A$ is the action space, $T : S \times A \times S \to [0, 1]$ is the transition probability function, $R : S \times A \to [0, R_{\max}]$ is the reward function, and $C : S \times A \to [0, C_{\max}]$ is the cost function. $\gamma \in [0, 1)$ is the discount factor, $\eta$ represents the initial state distribution, and $c_{th}$ is the cost threshold that sets the limit on cumulative costs for the policy. A policy $\pi : S \to \mathcal{P}(A)$ is a mapping from states to a probability distribution over actions, where $\pi(a|s)$ denotes the probability of selecting action $a$ in state $s$. Let the policy $\pi_\theta$ be parameterized by $\theta$. Given a policy $\pi$, its cumulative reward is defined as $R(\pi) = \mathbb{E}_{\tau \sim \pi} \left[ \sum_{t=0}^{\infty} \gamma^t r(s_t, a_t) \right]$, where $\tau = (s_0, a_0, s_1, a_1, \dots)$ is a trajectory induced by policy $\pi$, and the expectation is taken over the distribution of trajectories. Similarly, its cumulative cost is defined as $C(\pi) = \mathbb{E}_{\tau \sim \pi} \left[ \sum_{t=0}^{\infty} \gamma^t c(s_t, a_t) \right]$. The reward Q-function for a policy $\pi$ is defined as: $Q^\pi(s, a) = \mathbb{E}_{\tau \sim \pi} \left[ \sum_{t=0}^{\infty} \gamma^t r(s_t, a_t) \mid s_0 = s, a_0 = a \right]$. Similarly, the cost Q-function $Q_c^\pi(s, a)$ is defined as the expected cumulative cost starting from the same state-action pair $(s, a)$ and thereafter following policy $\pi$: $Q_c^\pi(s, a) = \mathbb{E}_{\tau \sim \pi} \left[ \sum_{t=0}^{\infty} \gamma^t c(s_t, a_t) \mid s_0 = s, a_0 = a \right]$. In CMDP, the goal is to find an optimal policy $\pi^*$ that maximizes $R(\pi)$, subject to the constraint that the cumulative cost $C(\pi)$ does not exceed a predefined threshold $c_{th}$. This can be formulated as the following:

$$\max_\pi R(\pi), \quad \text{s.t.} \quad C(\pi) \leq c_{th}. \tag{1}$$

To solve this, a common approach is to apply the Lagrangian relaxation method (Ray et al., 2019) by introducing a Lagrange multiplier $\lambda$ to enforce the cost constraint, leading to the following primal-dual formulation:

$$\min_{\lambda \geq 0} \max_\pi \left[ R(\pi) - \lambda(C(\pi) - c_{th}) \right], \tag{2}$$

which can be solved by iteratively updating policy $\pi$ and $\lambda$. $\lambda$ is updated with the learning rate $\alpha_\lambda$:

$$\lambda_{t+1} = \lambda_t + \alpha_\lambda(C(\pi_t) - c_{th}). \tag{3}$$

**Online Safe RL.** Primal-dual based algorithms have shown great effectiveness and superior performance in the literature for online safe RL. Without loss of generality, we consider SAC-lag (Ray et al., 2019) as the online algorithm, which integrates the widely used SAC algorithm (Haarnoja et al., 2018) with the Lagrange multiplier method. More specifically, SAC minimizes the following objectives for the actor (policy) and the

critic (Q-function), respectively:

$$\mathcal{L}_\pi^{SAC}(\theta) = \mathop{\mathbb{E}}_{s\sim d,\ a\sim\pi_\theta(\cdot|s)}[\alpha\log\pi_\theta(a|s) - Q(s,a;\omega)], \tag{4}$$

$$\mathcal{L}_Q^{SAC}(\omega) = \mathop{\mathbb{E}}_{(s,a,s')\sim d}[(\hat{Q}(s,a;\omega) - y(r,s'))^2] \tag{5}$$

where $y(r,s') = r + \gamma\mathbb{E}_{a'\sim\pi_\theta}[\hat{Q}(s,a';\omega') - \alpha\log\pi(a'|s')]$, $Q(s,a;\omega)$ is parameterized by $\omega$, $\hat{Q}(s,a;\omega')$ is the target reward Q-function parameterized by $\omega'$, $d$ represents the data distribution in the replay buffer, and $\alpha > 0$ is some constant. To be applied in online safe RL, SAC-lag adapts SAC by using the Lagrangian method, such that:

$$\mathcal{L}_\pi^{SAC}(\theta) = \mathbb{E}_{s\sim d}\mathbb{E}_{a\sim\pi_\theta(\cdot|s)}[\alpha\log\pi_\theta(a|s) - (Q(s,a) - \lambda Q_c(s,a))].$$

The optimization of the Q-functions for both reward and cost in SAC-lag is with Eq. 5 in SAC.

**Offline Safe RL.** Offline safe RL algorithms typically push up the cost estimations of OOD actions to avoid exploration beyond the offline dataset $\mathcal{D}$. Considering the comprehensive performance across various environments, in this paper we consider the SOTA Lagrangian-based algorithm for offline learning, namely CPQ (Xu et al., 2022). More specifically, CPQ first generates OOD actions via a conditional variational autoencoder (CVAE). The cost of the generated OOD actions is increased by minimizing the following loss function for cost critic ($Q_c$-function):

$$\mathcal{L}_{Q_c}^{CPQ}(\omega_c) = \mathbb{E}_D\Big[\big(Q_c(s,a;\omega_c) - \big(c + \gamma\mathbb{E}_{a'\sim\pi_\theta(\cdot|s')}\ [\hat{Q}_c(s',a';\omega'_c)]\big)\big)^2\Big] - \psi\mathbb{E}_{s\sim D, a\sim\nu}[Q_c(s,a;\omega_c)]$$

where $D$ represents distribution of $(s,a,s')$ in the offline dataset. $Q_c(s,a;\omega_c)$ is parameterized by $\omega_c$, $\hat{Q}_c(s,a;\omega'_c)$ is the target cost Q-function parameterized by $\omega'_c$, $\nu$ represents the distribution of OOD actions generated by the CVAE. Additionally, to ensure both constraint safety and in-distribution safety, CPQ updates the reward Q-function using only state-action pairs that satisfy the cost threshold $l$:

$$\mathcal{L}_Q^{CPQ}(\omega) = \mathbb{E}_D\big[(Q(s,a;\omega) - (r + \gamma\mathbb{E}_{a'\sim\pi_\theta(\cdot|s')}[\mathbb{I}(Q_c(s',a';\omega_c) < l)Q(s',a';\omega)]))^2\big]$$

where $\mathbb{I}(\cdot)$ is the indicator function. The policy loss function is defined as: $\mathcal{L}_\pi^{CPQ}(\theta) = -\mathbb{E}_D\big[\mathbb{E}_{a\sim\pi_\theta(\cdot|s)}[\mathbb{I}(Q_c(s,a;\omega_c) < l)Q(s,a,\omega)]\big]$. Similarly, when maximizing the reward, the policy only considers state-action pairs that meet the safety constraints. By assigning a higher cost to OOD actions, CPQ mitigates the OOD problem while meeting safety constraints.

**O2O Safe RL.** To the best of our knowledge, Guided Online Distillation (Li et al., 2024) is the only work studying O2O safe RL, which leverages a large-scale DT based on GPT-2 (Radford et al., 2019) as a guide policy to accelerate online learning, by following the idea of Jump-start RL (Uchendu et al., 2023). However, how to achieve fast safe online learning by finetuning a pretrained policy is still not clear due to the mismatch of Lagrange multipliers and erroneous Q-estimations from objective mismatch and offline cost sparsity.

In this work, we seek to enable faster and safer policy learning by finetuning policies and Q-functions pretrained via offline safe RL, using standard online safe RL methods. Our approach does not rely on large models and is intended as an initial step toward promoting further research on policy-finetuning-based O2O safe RL. In principle, any offline safe RL algorithm that outputs a policy and Q-functions can be used in the offline pretraining stage. More details of related work are provided in Section B.

## 3 O2O Safe RL with Value Pre-Alignment

As shown in Fig. 1, naively finetuning the offline policy for safe RL would not work and the finetuned policy shows clear "inertia" in improving its performance: within a long period after online finetuning starts, its cost stays far below the limit, but its reward is quite low and not improving. This implies that such a strategy automatically "inherits" the conservatism from offline safe RL and is reluctant to actively explore to fully utilize the safe gap below the cost limit. In this section, we delve into this failure of naive finetuning, which points to two unique challenges for policy finetuning in O2O safe RL, i.e., erroneous offline Q-estimations and Lagrange multiplier mismatch. To address these, we propose a framework for O2O safe RL, i.e., warM-stArt safe RL with Value prE-aLignment (Marvel).

### 3.1 Pre-Finetune Phase

*Challenge I: Erroneous Q-estimations resulting from objective mismatch and offline cost sparsity.* By learning from a fixed dataset only, offline safe RL typically suffers from large extrapolation errors for OOD actions beyond the support of the dataset. A general principle to handle this is to penalize the reward/cost estimations for the OOD actions in such a way that risky explorations outside the dataset are discouraged. The Q-functions can be optimized as follows:

$$Q: \ \min \mathbb{E}_D\left[\left(Q(s,a) - (r + \gamma \mathbb{E}_{a' \sim \pi_{\text{eval}}(\cdot|s')}[Q(s',a')])\right)^2\right] + \psi \cdot \mathcal{P}(s, a_{OOD}), \tag{6}$$

$$Q_c: \ \min \mathbb{E}_D\left[\left(Q_c(s,a) - (c + \gamma \mathbb{E}_{a' \sim \pi_{\text{eval}}(\cdot|s')}[Q_c(s',a')])\right)^2\right] - \psi_c \cdot \mathcal{P}_c(s, a_{OOD}). \tag{7}$$

Here $\pi_{\text{eval}}$ denotes policy under evaluation. $\psi \cdot \mathcal{P}(s, a_{OOD})$ and $\psi_c \cdot \mathcal{P}_c(s, a_{OOD})$ are the penalty terms. For instance, penalties are introduced in VOCE (Guan et al., 2024) to minimize the reward Q-values and maximize the cost Q-values for OOD actions. CPQ (Xu et al., 2022) increases the perceived cost of OOD actions during Q-function and policy updates. In contrast, the optimization of Q-functions in online safe RL is standard without any penalty terms, i.e, $\psi = 0$ in Eq. 6 and Eq. 7.

Obviously, Offline and online safe RL have distinct objectives for Q-functions, meaning pretrained Q-functions may not accurately estimate values for state-action pairs encountered during online interactions. As a result, offline policies tend to act overly conservatively, exploring only low-cost regions during online finetuning. This limitation actually arises from the joint effect of (i) sparse offline costs—leading the pretrained cost critic $Q_c$ to assign *low* costs to IND actions—and (ii) offline pessimism that assigns *high* costs to OOD actions through conservative extrapolation. Under a Lagrangian update, the policy thus prefers to stay within the low-cost IND region and avoids OOD regions that the critic deems high-cost, even when those regions contain high rewards. However, effective online learning requires identifying state–action pairs with both high rewards and low costs, which necessitates exploring areas with potentially higher costs. This objective mismatch is even more pronounced in O2O safe RL due to the additional cost Q-function.

*Solution: Value Pre-Alignment.* To address the first challenge, a naive approach is to reevaluate the offline policy $\pi_0$ in online environments using Monte Carlo simulations, which however introduces additional interaction costs. Motivated by the recent advances in Off-Policy Evaluation (OPE) (Uehara et al., 2022), we borrow the idea from Fitted Q Evaluation (Hao et al., 2021) to align the offline Q-functions with the online learning objectives for the offline policy, by using the offline dataset **before** online policy finetuning. Starting from the pretrained Q-functions, we seek to minimize:

$$\mathcal{L}_Q^{\text{VPA}}(\omega) = \mathbb{E}_D\left[\left(Q(s,a) - (\gamma \, \mathbb{E}_{a' \sim \pi_0(\cdot|s')}[\hat{Q}(s',a') - \alpha^{\text{VPA}} \log \pi_0(a'|s')] + r)\right)^2\right] \tag{8}$$

$$\mathcal{L}_{Q_c}^{\text{VPA}}(\omega_c) = \mathbb{E}_D\left[\left(Q_c(s,a) - (\gamma \, \mathbb{E}_{a' \sim \pi_0(\cdot|s')}[\hat{Q}_c(s',a') - \alpha_c^{\text{VPA}} \log \pi_0(a'|s')] + c)\right)^2\right] \tag{9}$$

where $\omega$ denotes parameters of Q function and $\hat{Q}$ is target Q function. Here the entropy terms can result in **both** higher rewards and costs for state-action pairs with high entropy, where pretrained policy is 'uncertain'. Offline policy remains **unchanged** during VPA to preserve knowledge extracted from offline data.

Assume that the mismatch between the offline policy-induced distributions of the learned policy $\pi_0$ and the offline behavior policy $\mu$ is bounded by constant $C < \infty$, i.e., $\max_{(s,a) \in S \times A} \frac{d^{\pi_0}(s,a)}{d^\mu(s,a)} \leq C$. Let $Q_0$, $Q_K$, and $\hat{Q}^{\pi_0}$ be Q-function obtained from offline phase, Q-function after $K$ iterations in VPA and the optimal (fixed-point solution) Q-function for reward of $\pi_0$, respectively (similarly, $Q_{c,0}$, $Q_{c,K}$, and $\hat{Q}_c^{\pi_0}$ for cost Q-functions). We can have following result to justify effectiveness of VPA:

**Theorem 1.** *Assume that the reward $r$ and cost $c$ are bounded by 1 for all $(s,a) \in S \times A$. With probability at least $1 - \delta$, the Q-estimation errors in VPA can be bounded in the weighted $L_2$-norm $\| \cdot \|_{2,d^{\pi_0}}$ under the*

*state-action distribution induced by policy $\pi_0$:*

$$\|Q_K - \hat{Q}^{\pi_0}\|_{2,d^{\pi_0}} \leq \frac{\sqrt{C\tilde{\epsilon}}}{1-\gamma} + \gamma^K \|Q_0 - \hat{Q}^{\pi_0}\|_{2,d^{\pi_0}}, \tag{10}$$

$$\|Q_{c,K} - \hat{Q}_c^{\pi_0}\|_{2,d^{\pi_0}} \leq \frac{\sqrt{C\tilde{\epsilon}_c}}{1-\gamma} + \gamma^K \|Q_{c,0} - \hat{Q}_c^{\pi_0}\|_{2,d^{\pi_0}} \tag{11}$$

*where $\tilde{\epsilon}$ and $\tilde{\epsilon}_c$ are the approximation errors. It decreases as the size of the dataset increases while increases with the values of $\alpha^{VPA}$ and $\alpha_c^{VPA}$, respectively.*

The full proof and the specific expression of $\tilde{\epsilon}$ are provided in Appendix H. As shown in Theorem 1, the overall estimation error consists of two terms, i.e., the statistical error in the first term and the iteration error in the second term. With a larger dataset and more expressive function approximator, the first term vanishes, and the overall error is dominated by the iteration error, which decays exponentially with training steps $K$. This indicates that VPA can yield increasingly accurate Q-function estimates. Moreover, since reward optimization dominates policy updates at the early stage of online finetuning when Lagrange multiplier is small, a larger uncertainty in reward Q-estimations is preferred to facilitate more online exploration. To this end, Theorem 1 further implies that a larger $\alpha^{VPA}$ should be selected in Eq. 8 for reward Q-values, whereas $\alpha_c^{VPA}$ in Eq. 9 should be smaller to ensure a more accurate estimation of cost Q-values. While the theory suggests improved Q estimations for IND and OOD actions, our empirical findings show estimations for OOD actions after VPA tends to be overoptimistic. This discrepancy is likely due to imperfect data coverage and function approximation errors. Interestingly, while such overestimation is problematic and avoided for offline RL without online explorations, it is indeed beneficial for O2O RL during early phase of online finetuning, which can encourage more active explorations beyond the offline dataset. While Luo et al. (2024) considered a similar analysis for unconstrained RL, we focus on the safe RL setting and further introduce VPA, a pre-finetuning procedure incorporating an entropy term to mitigate erroneous Q-value estimations.

Table 1: Spearman's rank correlation coefficients of $Q$-value and $Q_c$-value in *BallCircle* and *CarRun*. "Random" refers to rollouts starting from randomly initialized state-action pairs (potentially OOD), while "Dataset" refers to rollouts from state-action pairs sampled from the offline dataset.

| | VPA | BallCircle | | CarRun | |
|---|---|---|---|---|---|
| | | Random | Dataset | Random | Dataset |
| $Q$-value | Before | -0.2387 | -0.3852 | -0.1143 | -0.5078 |
| | After | 0.5661 | 0.8278 | -0.0125 | 0.8314 |
| $Q_c$-value | Before | -0.2521 | 0.1725 | -0.2431 | -0.4327 |
| | After | 0.3579 | 0.8252 | 0.1254 | 0.4937 |

To empirically characterize the performance of VPA in correcting Q-estimations, we leverage Spearman's rank correlation coefficient, which measures the strength and direction of a monotonic relationship between two ranked variables. The reason is that relative ranking of Q-values are more important than absolute values for policy update. We provide more details of Spearman's rank correlation coefficient in Appendix G.1. Specifically, given a dataset collected by the offline policy in the environment, we compare the ranking of learned reward/cost Q-values before and after VPA with that of estimated actual return via Monte Carlo simulations. A large Spearman's rank correlation coefficient implies that distribution of learned Q-values is more aligned with the distribution of true Q-values. As shown in Table 1, it is evident that coefficient increases significantly after VPA for both reward and cost Q-values, no matter if offline policy rolls out from a seen state-action pair in offline dataset or from a randomly selected OOD state-action pair. This clearly demonstrates effectiveness of VPA in aligning the pretrained Q-functions.

### 3.2  Finetune Phase

*Challenge II: Lagrange multiplier mismatch.*  Conventional value-based online safe RL relies on updating Lagrange multipliers alongside the policy and Q-functions during training, to push the overall cost below

the limit while balancing between maximizing the reward and minimizing constraint violations. Although the policy and Q-functions can benefit from offline pretraining for a warm start, offline safe RL algorithms like CPQ (Xu et al., 2022) and BEAR-lag (Ray et al., 2019) cannot accurately estimate Lagrange multipliers with regularizing strengths matching the cost of the offline policy, e.g., a small multiplier is not powerful enough to push down the policy cost, while a large multiplier prevents active exploration of high-reward state-action pairs. For instance, in BallCircle, the offline Lagrange multiplier value obtained using the BEAR-lag algorithm is approximately **1500**, whereas during online finetuning, the SAC-lag requires a value of only about **0.65**. The gap between these values precludes the direct use of offline Lagrange multipliers. Improper initialization can lead to extensive constraint violations or training stagnation, an issue we term as the **Lagrange multiplier mismatch**.

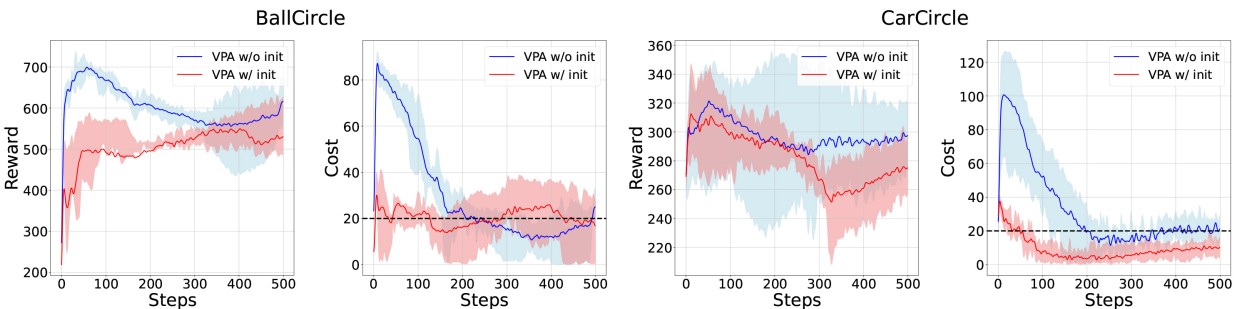

Figure 2: Comparison of online finetuning performance after VPA with two different initial values of the Lagrange multiplier. In 'VPA w/o init', the initial value is set to 0, whereas we initialize Lagrange multipliers with a good value found empirically (0.65 in BallCircle and 0.5 in CarRun) in 'VPA w/ init'. The multiplier is then updated using the standard dual ascent method.

On the other hand, as VPA promotes active exploration of high-reward state-actions by optimistically estimating rewards and pessimistically estimating costs, it inevitably increases the risk of exploring high cost state-actions, which in turn amplifies the need for appropriate Lagrange multipliers to quickly reduce the constraint violations.

Figure 2 compares online finetuning performance in BallCircle when using VPA with: (1) a well-initialized Lagrange multiplier vs. (2) zero initialization, both with standard dual ascent updates. A good initial multiplier enables effective cost control, while poor initialization leads to significant constraint violations and slower convergence. The results also imply that although VPA aligns the distributions of Q-values, it may introduce high costs for online finetuning, which can be addressed with an appropriate initial Lagrange multiplier.

*Solution: Adaptive PID Control.* Finding a good initial value of the Lagrange multiplier can address the mismatch problem and mitigate the risk brought by VPA. However, obtaining such a value by manual tuning is challenging, and no theory provides a reliable prediction. We therefore *adapt* the multiplier rapidly and effectively by using a PID-style controller that has shown strong empirical performance in online safe RL (Stooke et al., 2020; Yuan et al., 2022; Zhou et al., 2021), and we tailor it to the O2O setting.

We use a *discrete, position-form* PID update. Let the discounted total episode cost be $c_t$ at training iteration $t$, the limit be $c_{th}$, and the violation be $e_t = c_t - c_{th}$. Define an EMA-smoothed violation $\tilde{e}_t = a_p \tilde{e}_{t-1} + (1 - a_p)e_t$ with smoothing factor $a_p \in [0, 1)$. Define an EMA-smoothed cost $\tilde{c}_t = a_d \tilde{c}_{t-1} + (1 - a_d)c_t$ with smoothing factor $a_d \in [0, 1)$, and a derivative buffer of length $d \in \mathbb{N}$. The integral state and the derivative signal follow the implementation:

$$I_t = \max\big(0, \ I_{t-1} + K_i \, e_t\big), \qquad D_t = \max\big(0, \ \tilde{c}_t - \tilde{c}_{t-d}\big).$$

The multiplier is then

$$\lambda_t = \max\big(0, \ K_p \, \tilde{e}_t + I_t + K_d \, D_t\big). \tag{12}$$

This position-form matches the code path where the proportional term uses $\tilde{e}_t$, the integral term uses $e_t$, and the derivative term uses a lagged difference of smoothed costs $D_t$. The integral channel provides the same role as dual ascent.

VPA relaxes offline conservatism and increases exploration of high-return state–action pairs during early finetuning, which can raise violations. A larger control strength in early stages reduces violations rapidly, and a smaller control strength near the limit stabilizes learning and suppresses oscillations. This motivates an adaptive gain schedule.

Gains are adapted from a short buffer of smoothed costs. Let the buffer contain the last $n$ values of $\tilde{c}_t$ that are also used to form the derivative buffer. Define the average $\bar{c}_t = \frac{1}{n} \sum_{i=1}^{n} \tilde{c}_{t+1-i}$ and the standard deviation $\sigma_t = \sqrt{\frac{1}{n} \sum_{i=1}^{n} (\tilde{c}_{t+1-i} - \bar{c}_t)^2}$. With coefficients $\alpha, \beta, \gamma > 0$, the update is

$$K_p \leftarrow \text{clip}\left( K_p \cdot \left( 1 + \alpha \cdot \frac{\bar{c}_t - c_{th}}{\bar{c}_t} \right), \ K_{p,\min}, \ K_{p,\max} \right), \tag{13}$$

$$K_i \leftarrow \text{clip}\left( K_i \cdot \left( 1 + \beta \cdot \frac{\bar{c}_t - c_{th}}{\bar{c}_t} \right), \ K_{i,\min}, \ K_{i,\max} \right), \tag{14}$$

$$K_d \leftarrow \text{clip}\left( K_d \cdot \left( 1 + \gamma \cdot \frac{\sigma_t}{\bar{c}_t} \right), \ K_{d,\min}, \ K_{d,\max} \right). \tag{15}$$

This schedule increases $(K_p, K_i)$ when $\bar{c}_t > c_{th}$ to raise low-frequency gain and accelerate correction, and decreases $(K_p, K_i)$ when $\bar{c}_t < c_{th}$ to avoid overreaction and improve stability. The schedule increases $K_d$ when $\sigma_t$ is large to provide additional damping against short-horizon volatility. $K_d$ tends to plateau naturally once $\sigma_t$ stabilizes. The minimum and maximum bounds of each PID gain are set to 0.1 and 10 times its initial value, respectively. Since $K_d$ is non-decreasing by design, we set $K_{d,\min}$ only to keep notation consistent with $(K_{p,\min}, K_{i,\min})$.

Appendix D.3.2 demonstrates that these parameters are easy to tune but also robust in their own selection (e.g., same across all environments in the experiments).

In a nutshell, combining VPA and aPID leads to our proposed framework Marvel: 1) Given the pretrained policy and Q-functions from offline learning, Marvel first applies VPA to align the pretrained Q-functions for both reward and cost using the offline dataset; 2) Marvel next utilizes Lagrangian-based online safe RL algorithms to further finetune both the pretrained policy and aligned Q-functions, by using aPID to update the Lagrange multipliers. We present the algorithmic framework of Marvel in Appendix A.

Here VPA and aPID work in concert: aPID addresses the Lagrange multiplier mismatch problem and quickly pushes down the potential high cost resulted by VPA, whereas VPA facilitates active exploration of high-reward state-action pairs and the usage of the pretrained policy as a warm-start for fast online finetuning with aPID control.

## 4 Experiments

The objective of O2O safe RL is to utilize offline data to enable fast online finetuning, i.e., maximizing rewards while satisfying the cost threshold, with minimal interactions with the environment. Therefore, an important evaluation criterion for effective O2O safe RL methods is whether the method can **quickly** obtain a good policy with high rewards and low safety violations only after a limited number of online interactions, which is the setup considered in this paper. Due to the space limit, we delegate the experimental details and some additional results to Appendix D and Appendix G.

### 4.1 Evaluation Setup

*Benchmarks.* We consider the DSRL benchmark (Liu et al., 2023b) and select **ten** environments from the Bullet Safety Gym (Gronauer, 2022) and Safety Gymnasium (Ji et al., 2023): BallRun, BallCircle, CarRun, CarCircle, HalfCheetah, AntCircle, AntRun, DroneCircle, Hopper, and Swimmer (results for the last four are in Appendix D.1). The cost threshold is set to be 20 in these environments. As mentioned earlier in

Section 2, we choose CPQ and SAC-lag as base algorithms in our proposed framework Marvel for offline training and online finetuning, respectively, due to the effectiveness and representativeness of them. Each experiment was conducted using five random seeds, and the results were averaged to generate the final learning curves. We use a dataset that includes data provided by DSRL (Liu et al., 2024) and random data generated by a random policy to control the quality of the offline dataset.

*Baselines.* While Guided Online Distillation (Li et al., 2024) is the only work studying O2O safe RL, its usage of large pretrained model leads to an unfair comparison with standard RL frameworks using typically small-scale policy networks. In this work, we compare Marvel with **JSRL** (Uchendu et al., 2023), as Guided Online Distillation mainly follows this approach except using DT as the pretrained policy. Besides, we further adapt some SOTA approaches in O2O RL to O2O safe RL, including **SO2** (Zhang et al., 2024) and **PEX** (Zhang et al., 2023a), and a **Warm Start** approach as baselines. SO2 improves Q-value estimation through Perturbed Value Updates, JSRL and PEX utilize offline pretrained policies for exploration, and Warm Start directly finetunes the policy and Q-networks from offline safe RL without modifications. We compare with online learning from scratch, namely **From Scratch**, i.e., SAC-lag, which can achieve strong performance but require a large number of online interactions. For completeness, we also evaluate the performance of **Cal-QL** (Nakamoto et al., 2023) and **SAC-Lag with an offline replay buffer** across four representative environments: BallCircle, BallRun, CarCircle, and CarRun. The corresponding results are presented in Fig. 6. Given the interaction and network update settings in our experiments, we compare our implementation of SAC-lag with the standard FSRL library (Liu et al., 2024) in Appendix D.5, and the results show that their performance is highly comparable.

*More importantly, aPID is used to update Lagrange multipliers in all these baseline methods (instead of traditional dual ascent), which in fact already improves the performance of these methods compared to their original designs.* These baselines provide a meaningful comparison to demonstrate the effectiveness of Marvel, but also indicate the need of addressing the two challenges simultaneously for O2O safe RL. We provide the detailed descriptions of each baseline algorithm in Appendix C. Additionally, we provide a comparison with CDT (Liu et al., 2023c), using a smaller decision transformer (around 600k parameters) than the one in (Li et al., 2024), in Appendix D.1, and also demonstrate Marvel's superiority over baselines with large models.

Table 2: The "offline" column represents the performance of the offline safe RL pre-training. We present the performance at 20% of the total training steps during the finetuning process, as well as the performance after all the training steps are completed, e.g. "100 steps r/c" and "500 steps r/c" for reward and cost, respectively. Cost values greater than 20 are shown in gray. Bold numbers represent the highest reward while satisfying the cost threshold. The plots are in Appendix D.

| Environment | | Marvel (ours) | From Scratch | Warm Start | SO2 | JSRL | PEX | Offline |
|---|---|---|---|---|---|---|---|---|
| **BallCircle** | 100 steps: r/c | **580.3 / 19.2** | 170.8 / 88.9 | 54.7 / 0.0 | 54.4 / 0.0 | -7.6 / 137.5 | 0.0 / 119.7 | 166.0 / 11.0 |
| | 500 steps: r/c | **603.9 / 19.8** | 241.7 / 10.9 | 176.6 / 18.5 | 58.4 / 2.0 | 1.6 / 165.4 | -4.3 / 39.9 | |
| **BallRun** | 100 steps: r/c | 276.0 / 5.8 | 1055.9 / 75.4 | 346.3 / 22.1 | **338.6 / 19.9** | -120.5 / 37.2 | 270.3 / 42.9 | 262.0 / 3.0 |
| | 500 steps: r/c | 306.6 / 5.5 | **315.2 / 5.6** | 132.2 / 23.0 | 286.8 / 12.1 | 174.0 / 92.1 | -555.6 / 88.8 | |
| **CarCircle** | 100 steps: r/c | **330.6 / 18.8** | 49.1 / 16.4 | 73.2 / 71.1 | 69.3 / 17.6 | 2.4 / 36.3 | -0.5 / 69.4 | 265.0 / 14.0 |
| | 500 steps: r/c | **341.3 / 19.5** | 115.2 / 12.7 | 141.6 / 24.1 | 115.9 / 9.1 | 1.1 / 130.0 | -12.6 / 142.9 | |
| **CarRun** | 100 steps: r/c | **537.9 / 17.0** | 337.1 / 74.3 | 383.9 / 64.4 | 510.2 / 17.3 | -186.4 / 140.8 | -183.6 / 112.5 | 544.0 / 72.0 |
| | 500 steps: r/c | **547.5 / 18.2** | 293.5 / 7.2 | 391.4 / 16.9 | 522.2 / 16.2 | 126.6 / 128.2 | 153.0 / 38.8 | |
| **AntCircle** | 800 steps: r/c | **4.0 / 0.0** | 1.2 / 0.0 | 1.1 / 0.0 | 2.4 / 0.0 | 0.0 / 0.0 | 1.3 / 0.2 | 4.0 / 39.0 |
| | 4000 steps: r/c | 5.6 / 0.9 | 1.5 / 0.0 | 1.3 / 0.0 | 3.7 / 1.3 | 0.6 / 1.9 | **6.9 / 2.3** | |
| **HalfCheetah** | 800 steps: r/c | **1343.6 / 19.2** | 39.3 / 0.0 | 1695.9 / 180.5 | 1169.5 / 27.1 | 0.1 / 0.0 | -70.6 / 0.0 | 113.0 / 17.0 |
| | 4000 steps: r/c | **1544.8 / 20.0** | 1074.3 / 28.8 | 974.1 / 15.4 | 559.4 / 23.9 | 27.6 / 3.8 | -155.6 / 0.2 | |
| **BallCircle (BEAR-lag)** | 100 steps: r/c | **621.1 / 19.2** | 170.8 / 88.9 | 174.5 / 101.9 | -9.8 / 135.1 | 4.7 / 150.2 | 24.5 / 129.4 | 394.0 / 19.0 |
| | 500 steps: r/c | **552.3 / 15.2** | 241.7 / 10.9 | 440.2 / 100.0 | 71.8 / 16.6 | 10.5 / 163.0 | -13.6 / 5.8 | |
| **CarRun (BEAR-lag)** | 100 steps: r/c | 410.3 / 60.0 | 337.1 / 74.3 | 398.7 / 38.1 | **191.1 / 1.3** | 125.0 / 84.2 | 97.8 / 15.2 | 270.0 / 17.0 |
| | 500 steps: r/c | **510.5 / 19.2** | 293.5 / 7.2 | 204.6 / 50.1 | -59.6 / 0.0 | 31.9 / 113.6 | 200.6 / 34.0 | |

### 4.2 Main Results

As shown in Table 2, Marvel demonstrates better or comparable performance compared to all baselines consistently across all environments, i.e., achieving the higher return while keeping the cost below the threshold. In particular, as shown in Fig. 4, Marvel demonstrates superior stability compared to other baseline algorithms. Note that a key aspect of safe RL is ensuring safety, which requires keeping the cost below the predefined threshold while maximizing reward. Marvel achieves this balance effectively, as its cost remains below the threshold in all environments. In stark contrast, the naive warm start method proves largely ineffective, often causing performance drop or stagnation during training. Without aligning the Q-estimations, both JSRL and PEX struggles a lot to improve during online learning and fails to control the cost. While SO2 mitigates the inaccuracies of Q-estimations related to O2O RL, it does so only to a limited extent and cannot maintain its performance consistently across different environments, even though aPID has already been used to boost its performance. On the other hand, the fact that SO2 performs better than other baselines further indicates 1) the great potentials of enabling fast and safe online learning through policy finetuning (compared to using the pretrained policy only as a guide policy as in JSRL and PEX) and 2) the need of correcting pretrained Q-estimations before online finetuning.

More importantly, Marvel demonstrates the ability to find a good and safe online policy *very quickly* with minimal online interactions, as shown in Fig. 4 and Fig. 5 (shown in Appendix D). For instance, in the BallCircle and CarCircle environments, near-optimal performance can be achieved with merely **20 steps** of gradient updates. By leveraging offline policies and aligned Q-functions, Marvel rapidly explores high-reward regions while addressing the overly conservative nature of offline pretrained policies. Although initial finetuning may cause a cost spike (Fig. 4), aPID effectively reduces costs by guiding the policy towards low-cost state-actions in high-reward regions. Also, Marvel achieves the best max reward for a given cumulative cost, illustrated in Fig. 16, demonstrating superior performance while incurring fewer violations in the environment." Notably, the **same aPID parameters are used across all environments**, showcasing its robustness and adaptability.

**Compatibility of Marvel:** In O2O safe RL, compatibility with different offline safe RL methods is essential. Given the non-interactive nature of offline training and the potential unavailability of algorithms due to privacy concerns, this compatibility becomes even more critical compared to online algorithms. Our design of Marvel naturally fits a variety of offline safe RL methods and only requires a pretrained policy and Q-functions. To further verify this, the last two rows of table 2 show the training process using BEAR+Lagrangian (BEAR-lag) (Kumar et al., 2019) in the offline phase and SAC-lag in online finetuning. Note that BEAR-lag was not specifically designed for offline safe RL, but rather it incorporates the Lagrange multiplier into offline RL. In BallCircle, Marvel achieves the highest reward while satisfying cost constraints, and in the CarRun environment, it also outperformed others while maintaining cost below the threshold. This highlights the flexibility of our algorithm across different offline safe RL methods.

### 4.3 Ablation Studies

We conduct experiments in various setups to better understand Marvel. As shown in Fig. 3, the performance is best when both VPA and aPID are used. In contrast, if only VPA is used with traditional dual ascent during online finetuning, it significantly slows down safe online learning and takes a much longer time to reduce the cost. If only aPID is applied without VPA, the learning performance is very similar to naive policy finetuning, which struggles to improve due to the erroneous Q-estimations. We also evaluate the effectiveness of adaptive control in aPID, by comparing the performance between Marvel (VPA+aPID) and Marvel with aPID replaced by PID (VPA+PID). In Fig. 3, the training curve for cost exhibits significant fluctuations without using aPID. More critically, when the cost is close to the limit, PID cannot reduce its control strength. As a result, even if on average the cost of VPA+PID is close to the threshold, it is very frequent that the real-time cost exceeds the limit substantially, which is in fact not safe. aPID can adjust $\lambda$ more precisely, which directly affects how the policy balances reward and cost at every timestep. In contrast, a delayed or inaccurate adjustment of the Lagrange multiplier alters the relative weighting of reward and cost, thereby degrading reward performance, as observed in the CarCircle and HalfCheetah plots (Appendix D.2).

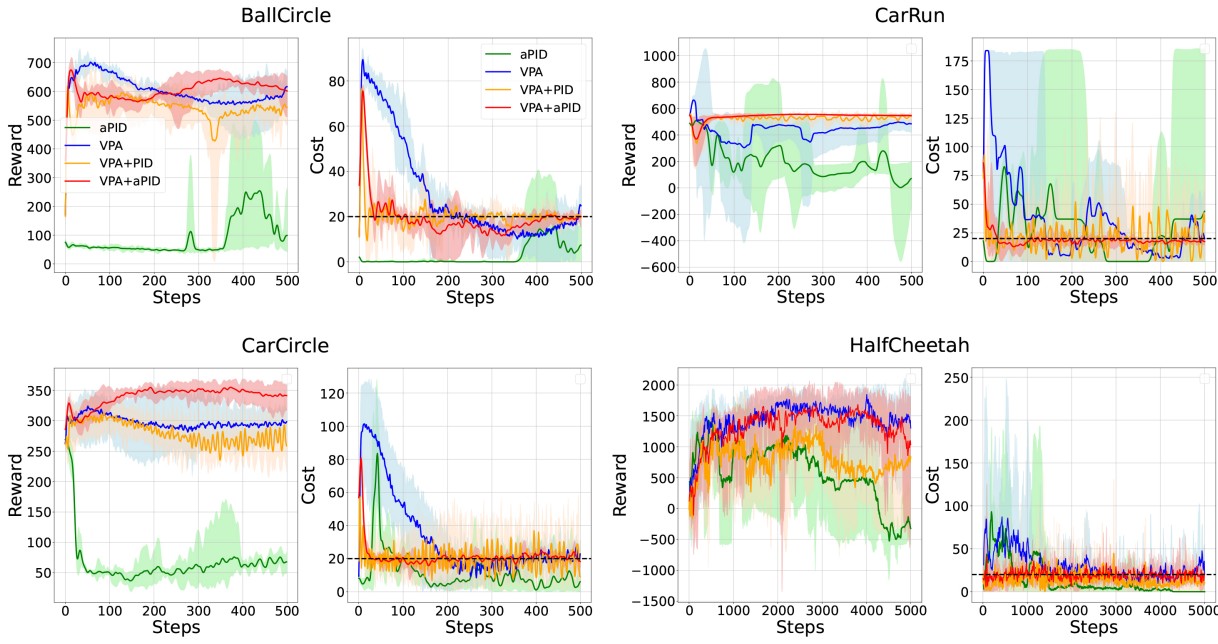

Figure 3: 'aPID' refers to adaptive PID applied only during online finetuning; 'VPA' refers to VPA is only used in pre-finetuning. 'VPA+PID' uses standard PID and VPA, while 'VPA+aPID' combines VPA with adaptive PID. VPA+aPID shows the best learning efficiency, stability, and convergence.

Table 3: Performance comparison between VPA+aPID (which is standard Marvel), VPA, and Warm Start when agent have different performance of offline policy. Note that although the second line shows Marvel exceeding the cost threshold, the violation is minor and can be corrected with just a few update steps.

| Environment | | VPA+aPID | VPA | Warm Start | Offline |
|---|---|---|---|---|---|
| **BallCircle** | 100 steps: r/c | **580.3 / 19.2** | 579.0 / 18.4 | 54.4 / 0.0 | 166.0 / 11.0 |
| | 500 steps: r/c | **610.1 / 19.6** | 511.3 / 16.9 | 176.6 / 18.5 | |
| | 100 steps: r/c | 543.6 / 20.5 | 576.8 / 92.1 | **430.7 / 13.5** | 446.0 / 6.0 |
| | 500 steps: r/c | 605.4 / 20.9 | 400.1 / 24.3 | **269.7 / 17.4** | |
| **CarCircle** | 100 steps: r/c | **283.8 / 20.0** | 320.1 / 49.7 | 107.3 / 8.8 | 145.0 / 17.0 |
| | 500 steps: r/c | **327.2 / 19.5** | 306.3 / 21.9 | 176.4 / 20.8 | |
| | 100 steps: r/c | **330.6 / 18.8** | 260.6 / 19.6 | 73.2 / 71.1 | 265.0 / 14.0 |
| | 500 steps: r/c | **341.3 / 19.5** | 283.1 / 17.2 | 141.6 / 24.1 | |

In practical O2O safe RL training, the quality of offline datasets can vary significantly, making it essential to evaluate whether Marvel maintains strong performance under different pretrained policies and Q-networks. As shown in Table 3, we train offline policies (and corresponding Q-networks) with varying quality levels. Marvel rapidly achieves near-optimal performance (with only a few online interaction steps, shown in Figure 9) regardless of the quality of the offline dataset or the effectiveness of the pretrained policy. This demonstrates the robustness of Marvel to variations in offline data quality. Furthermore, the algorithm consistently enforces cost constraints while improving rewards, thereby accommodating pretrained policies and Q-networks with diverse performance levels.

Regarding the necessity of each component of Marvel, Table 4 illustrates the importance of applying VPA to both the Q and Qc networks initialized from offline pre-trained models, as well as the contribution of the entropy term. Finetuning the pretrained Q-networks also outperforms learning new Q-networks from scratch

in VPA, which implies that the pretrained Q-networks, although not accurate, can still provide meaningful prior knowledge.

The training curves corresponding to Table 3 and Table 4 are presented in Appendix D.2, which further validating Marvel's robustness and design choices.: 1) Through extensive experiments across diverse environments, we confirm that the algorithm consistently maintains cost constraints while improving rewards across offline datasets of varying quality, which consequently accommodates offline pretrained policies and Q-networks with different performance levels. 2) The training dynamics confirm the rationality of VPA's design, particularly the necessity of updating all Q-networks in conjunction with the entropy term. 3) Parameter sensitivity analyses in Appendix D.3 and Appendix D.4 reveal acceptable performance variance despite the introduced hyper-parameters. Furthermore, we verify correctness of our baseline implementation by demonstrating comparable performance to the standard library FSRL (Liu et al., 2024) under identical experimental settings, including interaction steps and policy update frequencies.

| Offline | Q | Qc | Entropy | Pre-trained Q | Performance |
|---|---|---|---|---|---|
| | ✓ | ✓ | ✓ | ✓ | 603.9 / 19.8 |
| | ✓ | ✓ | | ✓ | 511.3 / 16.9 |
| 166.0 / 11.0 | ✓ | | | ✓ | 364.7 / 11.2 |
| | | ✓ | | ✓ | 64.1 / 5.3 |
| | ✓ | ✓ | ✓ | | 1.2 / 51.0 |

Table 4: Ablation study of VPA on BallCircle. "Offline" shows the initial policy/Q values. "Entropy" indicates whether entropy regularization is used. "Q"/"Qc" columns denote the targets of VPA. "Pre-trained Q" shows if VPA starts from an offline Q network. "Performance" reports final reward and cost.

## 5 Conclusion

O2O safe RL has great potential to put safe RL on the ground in real-world applications, by leveraging offline learning to facilitate fast online safe learning. In this paper, we proposed the first policy-finetuning based framework, namely Marvel, for O2O safe RL. In particular, by showing that naive finetuning would not work well, we identified two unique challenges in O2O safe RL, i.e., the erroneous Q-estimations and Lagrangian mismatch. To address these challenges, Marvel consisted of two key designs: 1) value pre-alignment to correct the Q-estimations before online finetuning, and 2) adaptive PID control to dynamically change the control parameters so as to rapidly and appropriately control the cost. Extensive experiments demonstrate the superiority of Marvel over multiple baselines. More importantly, Marvel is compatible to a variety of offline and online safe RL approaches, making it very practically appealing. We hope our work bridges the gap between offline and online safe RL, distinct from unconstrained RL, and enhances the efficiency of online safe RL, paving the way for practical applications.

## Acknowledgement

We thank the anonymous reviewers and the Action Editor for their constructive feedback and helpful suggestions, which greatly improved the quality and clarity of this paper.

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

# A  Overview of Algorithm

---

**Algorithm 1** Marvel

---

1: **Input:** Offline dataset $\mathcal{D}_{off}$, environment $\mathcal{E}$; offline RL loss functions $\{L_{off}^{Q_\phi}, L_{off}^{Q_{c_{\phi_c}}}, L_{off}^{\pi_\theta}\}$, and online RL loss functions $\{L_{on}^{Q_\phi}, L_{on}^{Q_{c_{\phi_c}}}, L_{on}^{\pi_\theta}\}$; offline learning rates $(\eta_Q^{off}, \eta_{Q_c}^{off}, \eta_\pi^{off})$; online learning rates $(\eta_Q^{on}, \eta_{Q_c}^{on}, \eta_\pi^{on})$; PID hyper-parameters and initial controller state $(\lambda_0, I_0, D_0)$.

2: *% These loss functions define a generic actor–critic safe RL algorithm: $L^{Q_\phi}$ and $L^{Q_{c_{\phi_c}}}$ are the objective functions for learning the reward critic $Q$ and the cost critic $Q_c$, respectively; $L^{\pi_\theta}$ is the policy optimization objective. Marvel treats them as modular components, allowing any safe RL algorithm to be plugged into the framework.*

3: *% In our implementation, we instantiate the offline losses using CPQ (Xu et al., 2022) and the online losses using SAC-lag (Ray et al., 2019), but Marvel is compatible with any actor–critic safe RL formulation.*

4: **Output:** Trained policy parameters $\theta$, critic parameters $\phi, \phi_c$, online replay buffer $\mathcal{D}_{on}$, and final Lagrange multiplier $\lambda$.

5: Initialize network parameters $\phi, \phi_c, \theta$; initialize $\mathcal{D}_{on} \leftarrow \emptyset$; set $(\lambda, I, D) \leftarrow (\lambda_0, I_0, D_0)$.

6: **while** in offline training phase **do**

7:    Sample minibatch $(s, a, r, c, s') \sim \mathcal{D}_{off}$

8:    $\phi \leftarrow \phi - \eta_Q^{off} \nabla_\phi L_{off}^{Q_\phi}$

9:    $\phi_c \leftarrow \phi_c - \eta_{Q_c}^{off} \nabla_{\phi_c} L_{off}^{Q_{c_{\phi_c}}}$

10:    $\theta \leftarrow \theta - \eta_\pi^{off} \nabla_\theta L_{off}^{\pi_\theta}$

11: **end while**

12: *% VPA with offline dataset*

13: **while** in VPA phase **do**

14:    **for** each VPA step **do**

15:       Sample transitions $(s, a, r, c, s') \sim \mathcal{D}_{off}$

16:       Update $Q$ by Eq. 8, update $Q_c$ by Eq. 9

17:    **end for**

18: **end while**

19: **while** in online training phase **do**

20:    Roll out one episode in $\mathcal{E}$ with policy $\pi_\theta$, collect transitions $(s_i, a_i, r_i, c_i, s_{i+1})$ and append to $\mathcal{D}_{on}$

21:    Compute total episode cost $c_t$ and update EMA-smoothed cost $\tilde{c}_t$

22:    **for** each update step **do**

23:       Sample minibatch $(s, a, r, c, s') \sim \mathcal{D}_{on}$

24:       $\phi \leftarrow \phi - \eta_Q^{on} \nabla_\phi L_{on}^{Q_\phi}$

25:       $\phi_c \leftarrow \phi_c - \eta_{Q_c}^{on} \nabla_{\phi_c} L_{on}^{Q_{c_{\phi_c}}}$

26:       $\theta \leftarrow \theta - \eta_\pi^{on} \nabla_\theta L_{on}^{\pi_\theta}$

27:       Update $\lambda$ by Eq. 12 with input $\tilde{c}_t$

28:       Update PID parameters by Eq. 15

29:    **end for**

30: **end while**

---

# B  Related Work

**Online Safe RL.**  Online safe RL approaches can be generally divided into several categories. The first category includes primal-dual based methods, such as PDO (Chow et al., 2018a), which combines PPO (Schulman et al., 2017) with the Lagrange multiplier method to obtain a policy that satisfies safety constraints. CPPO-PID (Stooke et al., 2020) combines PID control with Lagrangian methods to dampen cost oscillations. Similar Lagrangian-based methods are applied in conjunction with other unconstrained safe RL algorithms, such as TRPO-lag, PPO-lag, and SAC-lag. CPO (Achiam et al., 2017) inherits from TRPO

(Schulman, 2015), optimizing with the Lagrange multiplier method within the trust region. CUP (Yang et al., 2022) extends CPO by incorporating the generalized advantage estimator. In comparison, RCPO (Tessler et al., 2018) uses different update rates for the primal and dual variables. Two-stage iterative methods have also been developed for online safe RL, e.g., PCPO (Yang et al., 2020) and FOCOPS (Zhang et al., 2020). Besides the primal-dual based methods, primal methods, which are also known as Lyapunov methods, have been leveraged in some studies for online safe RL. For instance, IPO (Liu et al., 2020) uses logarithmic barrier functions. P3O (Zhang et al., 2022) employs an exact penalty function to derive an equivalent unconstrained objective and restrict policy updates within the trust region. (Chow et al., 2018b) leverages Lyapunov functions to handle constraints, which contains two parts, safe policy iteration and safe value iteration. Additionally, some studies (Wabersich et al., 2023; Choi et al., 2020) borrow techniques from the control theory, such as HJ reachability (Bansal et al., 2017; Yu et al., 2022) and control barrier functions (Ames et al., 2019), to ensure state-wise zero costs.

**Offline Safe RL.** Offline safe RL seeks to learn a safe policy from static datasets without online environmental interactions. Similar to online safe RL, Lagrangian methods can still be applied here, by adapting offline unconstrained RL algorithms like BCQ (Fujimoto et al., 2019) and BEAR (Kumar et al., 2019) to the safe RL setting. CPQ (Xu et al., 2022) uses a VAE to detect OOD (Ren et al., 2019) actions and penalizes them in terms of cost. COptiDICE (Lee et al., 2022a) extends OptiDICE (Lee et al., 2021) by adding safety constraints and derives a safe policy through the stationary distribution of the optimal policy. FISOR (Zheng et al., 2024) decouples the process of satisfying safety constraints from maximizing rewards and employs a diffusion model as the policy. VOCE (Guan et al., 2024) estimates Q-values of both cost and reward in a pessimistic way, mitigating extrapolation errors caused by OOD actions. Decision transformer (DT) (Chen et al., 2021) has also been applied to safe RL, leading to constrained decision transformer (Liu et al., 2023c).

**O2O Unconstrained RL.** O2O RL has recently attracted much attention in the unconstrained case, where a policy pretrained on an offline dataset is used to assist online policy learning, e.g., through finetuning or serving as a guide policy. More specifically, (Hester et al., 2018; Nair et al., 2018; Rajeswaran et al., 2017) and (Rudner et al., 2021) explore various combinations of offline demonstration data with online learning. The core idea is that pure offline RL often struggles with limited performance due to heavy reliance on dataset quality. However, if interaction with the environment is allowed, pffline pretrained policy can be finetuned for improved performance. However, naive implementation of this process often leads to suboptimal performance (Nair et al., 2020; Uchendu et al., 2023). AWAC (Nair et al., 2020) prioritizes actions with high advantage estimates, while AW-Opt (Lu et al., 2022) builds on AWAC by applying positive sample filtering and using hybrid actor-critic exploration during online finetuning. (Lee et al., 2022b) finetunes the pretrained policy by balancing the offline and online datasets. FamO2O (Wang et al., 2024) trains a family of policies using a universal model and then employs a balance model to select the most suitable policy for each state. Cal-QL (Nakamoto et al., 2024) constrains the updates to the Q-network during online finetuning to prevent underestimation of the Q-values. SO2 (Zhang et al., 2024) improves Q-value estimation by updating Q-values more frequently and using noise-augmented actions. Instead of directly finetuning the pretrained policy, Jump-start RL (Uchendu et al., 2023) and PEX (Zhang et al., 2023a) follows another direction to leverage the offline policy, by using it to guide the update of the online policy during online learning. Zhou et al. (2024) propose to finetune an offline policy without retaining the offline data. Their method first warms up training by interacting with the environment using the offline policy before performing any policy updates.

## C  Details on Baselines

Considering the characteristics of safe RL, which requires keeping the cost below a certain threshold, not all O2O unconstrained RL algorithms are suitable for O2O safe RL. For instance, AWAC (Nair et al., 2020), which maximizes the advantage function, has not yet been applied in the safe RL context. We compare Marvel with the following baselines:

**SO2** (Zhang et al., 2024). By analyzing Q-value estimation in offline to online transitions, the SO2 algorithm achieves more accurate Q-value estimation through Perturbed Value Update and by increasing the frequency of Q-value updates.

**JSRL** (Uchendu et al., 2023). JSRL employs an offline pretrained policy as the exploration policy and a policy under training during the online phase as the target policy. Initially, the exploration policy is used, followed by the target policy during online interaction to facilitate curriculum learning. To adapt to the safe RL setting, we update the Lagrange multipliers using the aPID method when updating the target policy.

**PEX** (Zhang et al., 2023a). Similar to JSRL, PEX uses an offline pretrained policy and a policy under training during the online phase for online interaction. However, PEX selects one of the actions based on the Q-networks's value estimation of actions chosen by the two policies. To meet the safe RL requirements concerning cost, like the modifications to JSRL, we use the aPID method to update the Lagrange multipliers.

**Cal-QL** (Nakamoto et al., 2023). This method fine-tunes policies using a better-calibrated Q-function. We employ the aPID method to update the Lagrange multipliers.

**SAC with an offline replay buffer**. This baseline naively incorporates offline data into the online replay buffer. As with our modifications before, we use the aPID method to update the Lagrange multipliers.

**Warm Start**. We directly utilize the policy, Q-network, and Qc-network networks obtained from offline safe RL without any modifications (no VPA and aPID), and apply online safe RL algorithms for finetuning.

We selected SO2, JSRL, PEX and Warm Start as baselines because they represent prominent methods in O2O RL, and adapting them to the safe RL context provides a meaningful comparison. Including these baselines allows us to demonstrate the effectiveness of Marvel in a fair and relevant context.

# D    More Experimental Results

## D.1    More experiments

In Fig. 4, we provide training curve of Marvel and baseline algorithms in BallRun, BallCircle, CarRun, CarCircle, HalfCheetah, AntCircle. In environments like BallCircle and CarCircle, Marvel finds a good policy within less than 15 steps, dominating baseline methods in both performance and speed. *Note that all approaches indeed start from the offline policy and Q-functions, i.e., the same point at step 0 (ignored in all figures).*

In Fig. 5, we additionally provide experimental results in more environments, including DroneCircle, AntRun, Hopper, and Swimmer. The results indicate that our proposed Marvel algorithm achieves competitive performance across these settings.

In Fig. 6, **Cal-QL**, which fine-tunes policies using a better-calibrated Q-function, demonstrates reasonable performance. However, since it was not originally designed for safe reinforcement learning, its direct application in this setting is inappropriate, since it fails to fully address the issues of erroneous Q-estimations and Lagrangian mismatch, as illustrated before. Similarly, the naive use of offline data, **SAC** with an offline replay buffer, fails to achieve satisfactory outcomes, as its incurred cost substantially exceeds the predefined threshold. This violation of safety constraints highlights the algorithm's inability to succeed under such conditions.

In Table 5, We conpare our Marvel with CDT(Liu et al., 2023c). Due to CDT's design for offline settings, we use the replay buffer to apply CDT in the online setting, which is represented as "CDT-finetune" in the table. It can be seen that CDT, having more parameters, generally performs better than CPQ in the offline phase. However, since the finetuning process was not specifically designed, performance stagnation and even degradation occur during finetuning.

## D.2    More ablations

### D.2.1    VPA on Q vs Qc vs Both and if VPA need entropy term

Fig. 7 presents more detailed ablation experiments, including whether VPA needs to be applied to both the Q-network and Qc-network, as well as whether the entropy term should be added to VPA. By comparing VPA(Q), VPA(Qc), and VPA(Q+Qc), we can observe that applying VPA solely to the Qc-network results

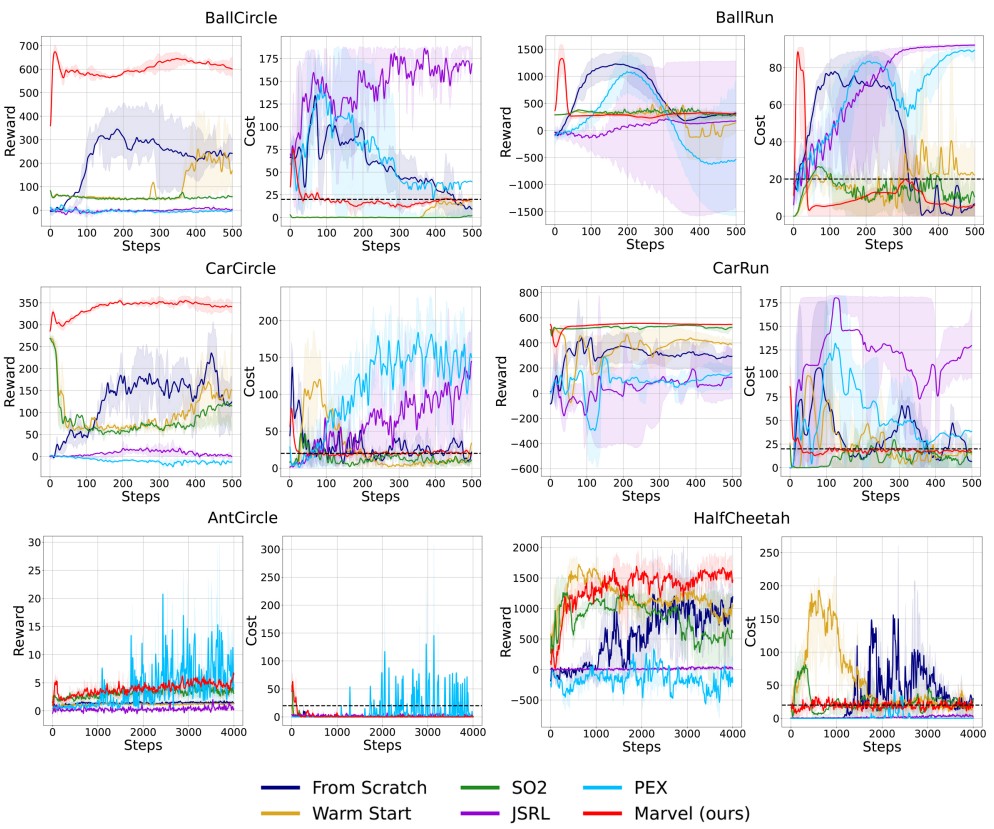

Figure 4: Performance comparison between Marvel and baseline methods in multiple environments. It is clear that Marvel can quickly find a high-return policy while keeping the cost below the limit.

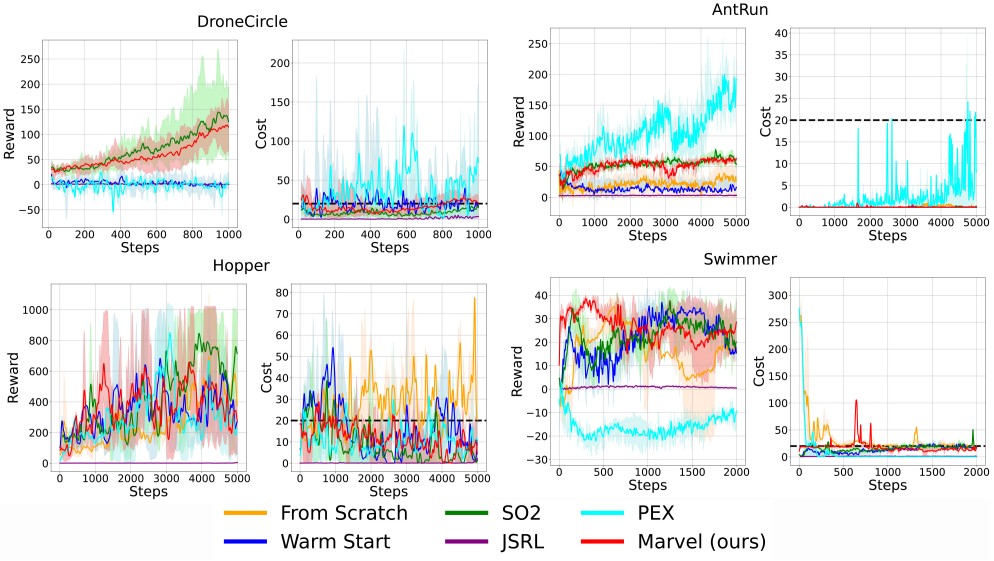

Figure 5: We provide experiments on more environments.

in very poor performance during online finetuning. For example, in the BallCircle environment, results similar to naive finetuning shown in Fig. 1 were observed. On the other hand, applying VPA only to the Q-network leads to significant instability during finetuning (e.g., large error bands in BallCircle, CarCircle,

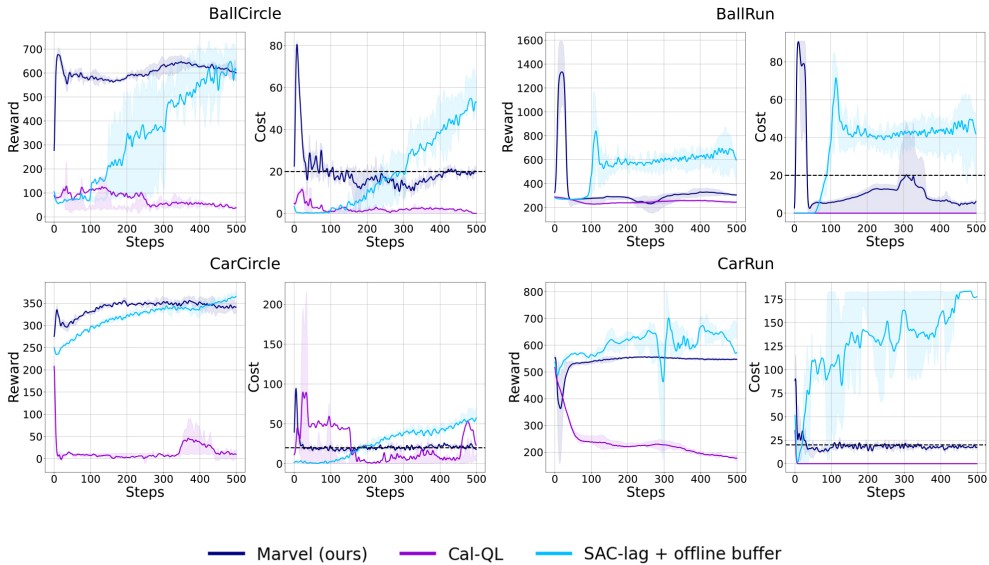

Figure 6: This figure presents a performance comparison between Marvel, Cal-QL, and SAC with an offline replay buffer. As shown, Marvel achieves the best overall performance.

Table 5: Comparison of Marvel and CDT

| Environment | Offline | | Online | |
|---|---|---|---|---|
| | **CPQ** | **CDT** | **Marvel (ours)** | **CDT-finetune** |
| **BallCircle** | 166.0 / 11.0 | 521.7 / 18.3 | 603.9 / 19.8 | 468.7 / 9.3 |
| **BallRun** | 262.0 / 3.0 | 448.5 / 48.5 | 306.6 / 5.5 | 449.0 / 50.3 |
| **CarCircle** | 265.0 / 14.0 | 328.7 / 18.1 | 341.3 / 19.5 | 320.2 / 23.1 |
| **CarRun** | 544.0 / 72.0 | 553.9 / 25.1 | 547.5 / 18.2 | 542.0 / 23.3 |
| **AntCircle** | 4.0 / 39.0 | 221.6 / 54.7 | 5.6 / 0.9 | 161.2 / 36.4 |
| **HalfCheetah** | 113.0 / 17.0 | 1358.0 / 24.3 | 1544.8 / 20.0 | 1536.7 / 3.9 |

and HalfCheetah) and poor performance in terms of cost (e.g., the cost curve in CarRun shows a sharp increase beyond the cost threshold). This occurs because if only the reward is optimistically estimated while the cost is pessimistically overestimated, it causes the agent to neglect the cost during exploration, adversely affecting finetuning performance. The experiments demonstrate that applying VPA to both the Q-network and Qc-network simultaneously has the best results, which aligns with the motivation discussed in Section 3. Comparing VPA(Q+Qc) with VPA(Q+Qc) with entropy, it is evident that optimistically estimating both reward and cost, while aligning with the pretrained policy, proves to be effective.

### D.2.2 Finetune Q-networks vs train new Q-networks in VPA

In Marvel, VPA finetunes the offline pretrained Q-networks. Fig. 8 illustrates the training curves when, instead of finetuning the pretrained Q-networks, the Q-networks are retrained from scratch during the VPA phase and subsequently finetuned online. As shown, finetuning the pretrained Q-networks achieves better performance. This is because, although the pretrained Q functions may be inaccurate, they still provide meaningful prior knowledge from the offline dataset and serve as a valuable starting point for Q function finetuning. This would generally speed up the learning and lead to a better local optima compared to learning from scratch based on the offline data from a random initial point. Moreover, considering the limited number

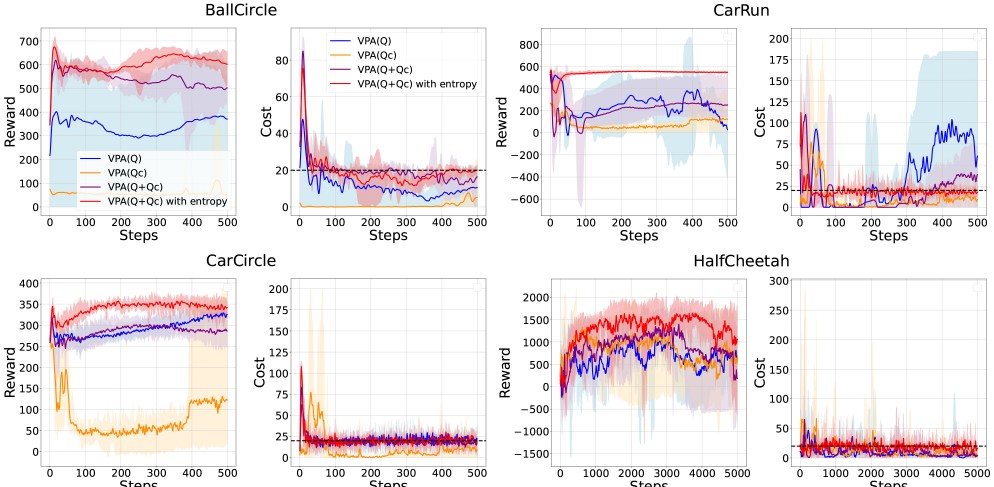

Figure 7: In the figure, VPA(Q), VPA(Qc), and VPA(Q+Qc) represent applying VPA to the Q-network, the Qc-network, and both simultaneously, without using the entropy term. This corresponds to setting $\alpha$ and $\alpha_c$ to 0 in Eq. 8 and Eq. 9. Conversely, VPA(Q+Qc) with entropy indicates that the entropy term is used in VPA, meaning $\alpha_c$ and $\alpha_c$ are non-zero. In all experiments represented by the curves, we employed aPID.

of steps allowed in VPA for efficiency, directly learning completely forgoes the knowledge learned offline and can fail to find good Q estimations.

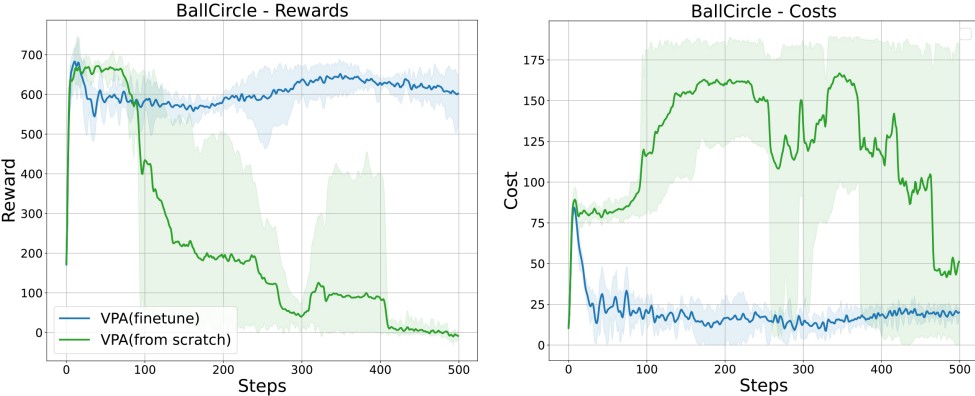

Figure 8: In the figure, "VPA (finetune)" refers to finetuning the offline pretrained Q-networks during the VPA phase, while "VPA (from scratch)" refers to training new Q-networks from scratch during the VPA phase.

### D.2.3 Marvel's Ability to Improve Pretrained Policies at Different Performance Levels

As shown in Fig. 9, regardless of the quality of the offline dataset or the performance of the pretrained policy, Marvel is able to quickly achieve optimal performance with only a few online interaction steps. This highlights the robustness of the Marvel algorithm to variations in the quality of the offline dataset.

### D.3 Parameter sensitivity of aPID

### D.3.1 PID

The SAC-lag algorithm in (Liu et al., 2024) utilizes PID control, with PID parameters carefully optimized. However, if their provided parameters are used directly under the environmental settings, policy updates,

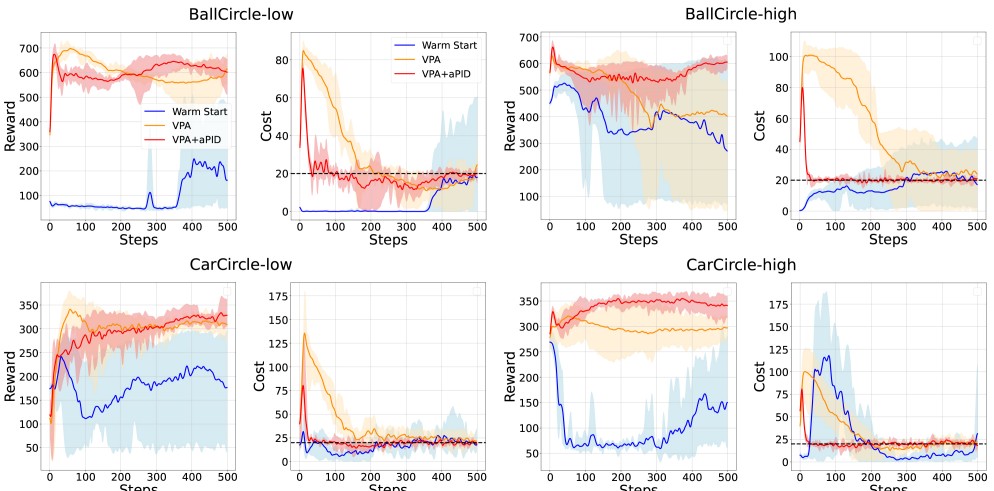

Figure 9: In the figure, "high" and "low" represent the different performance levels of offline pretrained policies resulting from varying quality in the offline dataset. These policies are then finetuned online. The results demonstrate that the Marvel algorithm is robust to both different offline dataset qualities and pretrained policy performances.

and Q-network update configurations of this paper, the performance is suboptimal. Fig. 10 presents a comparison, showing that when the PID parameters from the FSRL library are applied, the performance of online finetuning is significantly degraded. It is clear that the implementation of PID in our paper indeed significantly outperforms the implementation of PID provided by FSRL. More importantly, even with inappropriate PID parameters, aPID effectively boosts performance, achieving higher rewards while maintaining more stable cost levels.

### D.3.2 Parameters in aPID

$\alpha$, $\beta$, and $\gamma$ are the parameters used in aPID to adjust the PID parameters. These parameters enhance the robustness of the initial settings for the PID parameters while being inherently robust themselves. Although our method aPID introduces more parameters, this is very common for adaptive algorithms in order to control the adaptation during the learning procedure. Fig. 11 illustrates the performance under various combinations of $\alpha$, $\beta$, and $\gamma$, with values ranging from 0.01 to 0.5. All curves achieve similar performance in terms of reward and cost by the end of training. This demonstrates that these parameters are both easy to tune and robust in their selection.

### D.4 Parameter sensitivity of VPA

The selection of $\alpha$ and $\alpha_c$ follows a similar approach to the selection of $\alpha$ in SAC. These values need to be empirically determined based on the evaluation results of the pretrained policy, the entropy of the policy, and the scale of the Q-values provided by the Q networks. It is crucial to ensure that the values of these parameters do not cause the entropy term to dominate the Q-value update process. The tuning process involves starting with small values, such as those in the range of $1 \times 10^{-5}$. Considering that the entropy value is typically a negative single-digit number, the upper limit for $\alpha$ and $\alpha_c$ should generally be around $1 \times 10^{-1}$. For relatively conservative offline pretrained policies, larger values of $\alpha$ and $\alpha_c$ may be more suitable.

To demonstrate the robustness of the chosen $\alpha$ and $\alpha_c$, we scaled the values provided in this paper by a factor of five, ranging from $1 \times 10^{-4}$ to $3 \times 10^{-5}$. As shown in Fig. 12, the choice of different $\alpha$ and $\alpha_c$ values has minimal impact on the final performance.

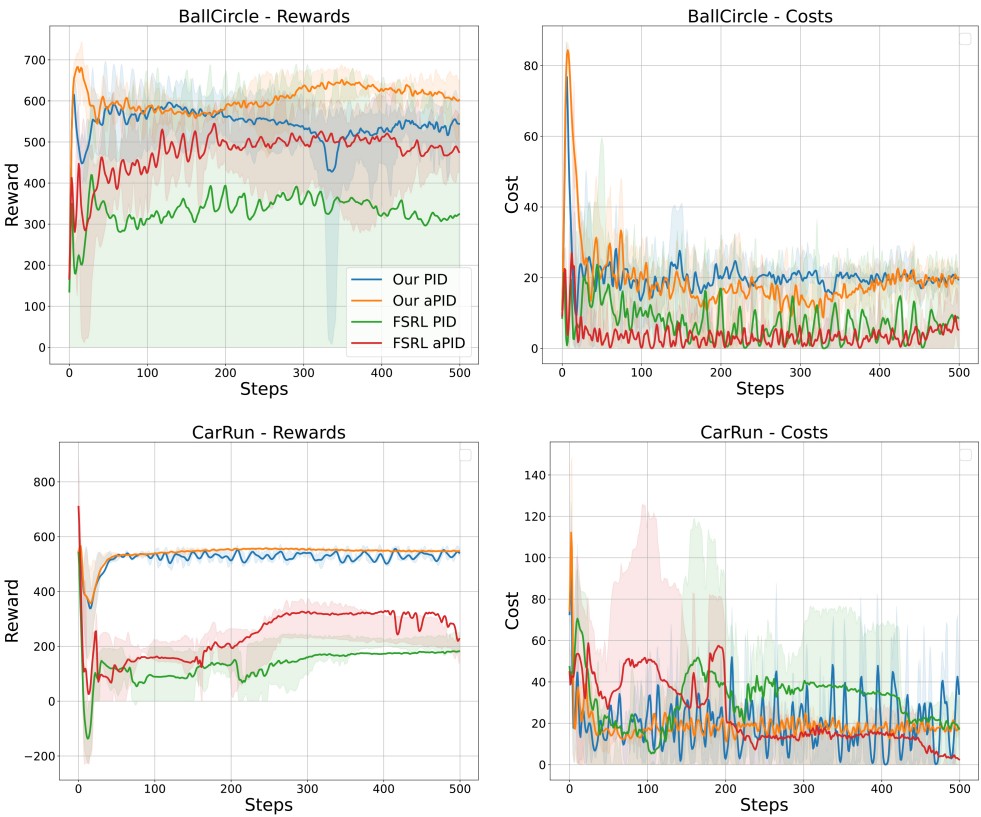

Figure 10: "Our PID" and "Our aPID" refer to using the PID and aPID parameters proposed in this paper for adjusting the Lagrange multipliers, respectively. Similarly, "FSRL PID" and "FSRL aPID" represent the parameters provided by the FSRL library for the same purpose.

### D.5 Correctness of our implementation of SAC-lag

The primary goal of O2O safe RL algorithms is to achieve competitive performance with **minimal environment interactions** and in the **shortest time** by leveraging offline information to accelerate online learning. In contrast, the algorithm in (Liu et al., 2024) (and other similar online algorithms) achieves higher performance but relies on **significantly more interactions**. For example, in the BallCircle environment, (Liu et al., 2024) utilized 1.5 million environment interactions, whereas our method required only 120,000 interactions (with an average of 600 interactions per gradient update). This significant reduction highlights the efficiency of our approach, particularly in resource-constrained and safety-critical settings where the number of online interactions is strictly limited.

To validate the correctness of our implementation of SAC-lag, we conducted additional experiments comparing it to the SAC-lag implementation provided by the FSRL library under the same experimental settings (including both environment interaction steps and policy update frequencies). The results, presented in Fig. 13, show that both implementations demonstrate similar performance in terms of reward and cost. This validates the correctness of our implementation and ensures its reliability as a baseline for comparisons in our study.

### D.6 Without VPA but with good initial values of the Lagrangian multipliers

As shown in Fig. 14, when VPA is not used and the Lagrange multipliers are updated using dual ascent (as described in Eq. 3), even with appropriately chosen initial values for the Lagrange multipliers ("Warm Start

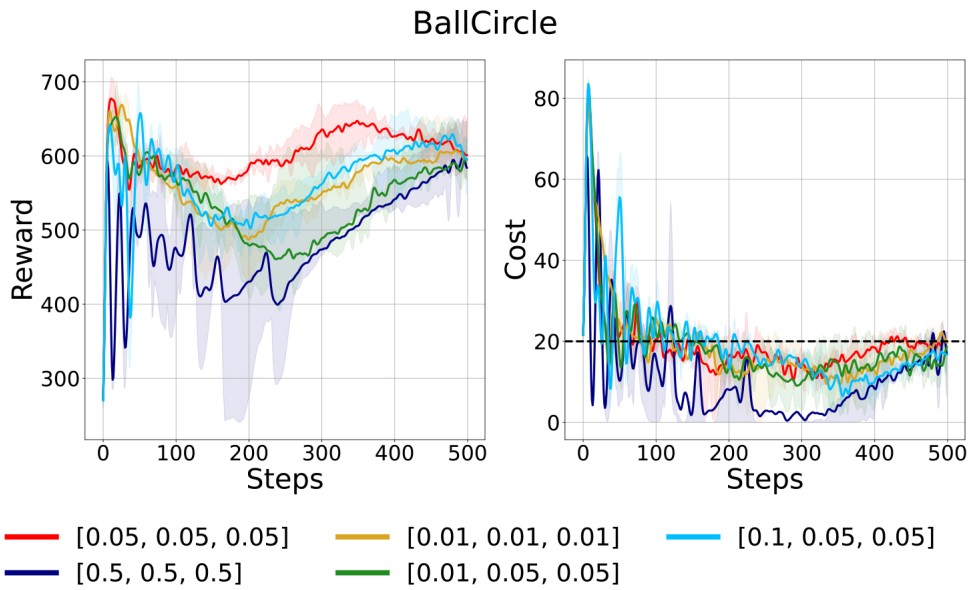

Figure 11: The three numbers within "[ ]" represent the three parameters for adaptively adjusting $K_p$, $K_i$ and $K_d$, namely $\alpha$, $\beta$ and $\gamma$.

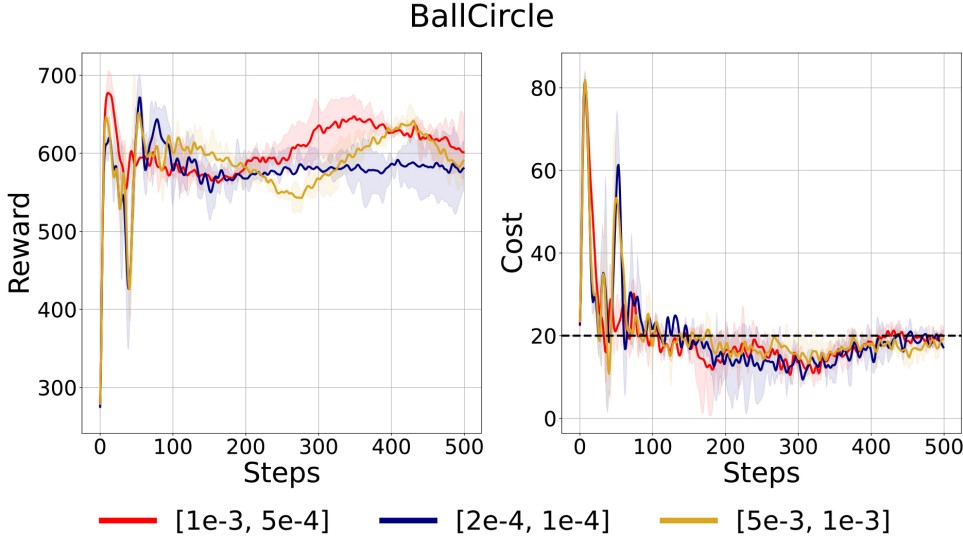

Figure 12: The two numbers inside "[ ]" represent the values of $\alpha$ and $\alpha_c$ used in VPA, as described in Eq. 8 and Eq. 9.

w/ lag init"), the performance, while better than initializing with zero ("Warm Start"), still falls short of achieving optimal results.

## D.7   Additional Baseline: Distilling Q-values of the Behavior Policy

To further evaluate the effectiveness of Marvel, we include an additional baseline that distills the Q-values of the policy which generated the offline dataset. In offline RL, the dataset is not necessarily produced by a single policy; it can be a mixture of trajectories collected from multiple behavior policies of varying quality. To align with such a mixed dataset, we first use behavior cloning (BC) to obtain a policy consistent with the

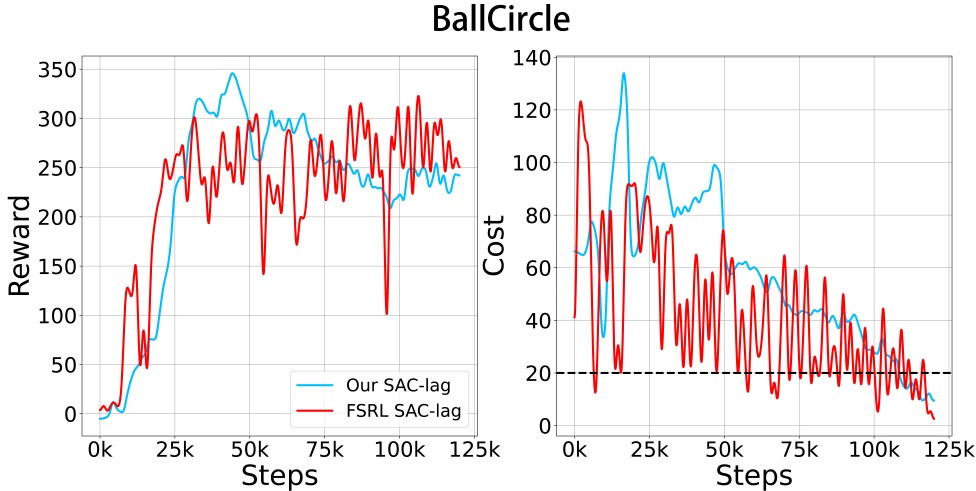

Figure 13: "Our SAC-lag" refers to the SAC-lag algorithm implemented in this paper, while "FSRL SAC-lag" represents the SAC-lag algorithm provided by the FSRL library. Using the same environment settings (including interaction steps) and update frequencies as in this paper, the results from the FSRL library are shown to be similar to ours.

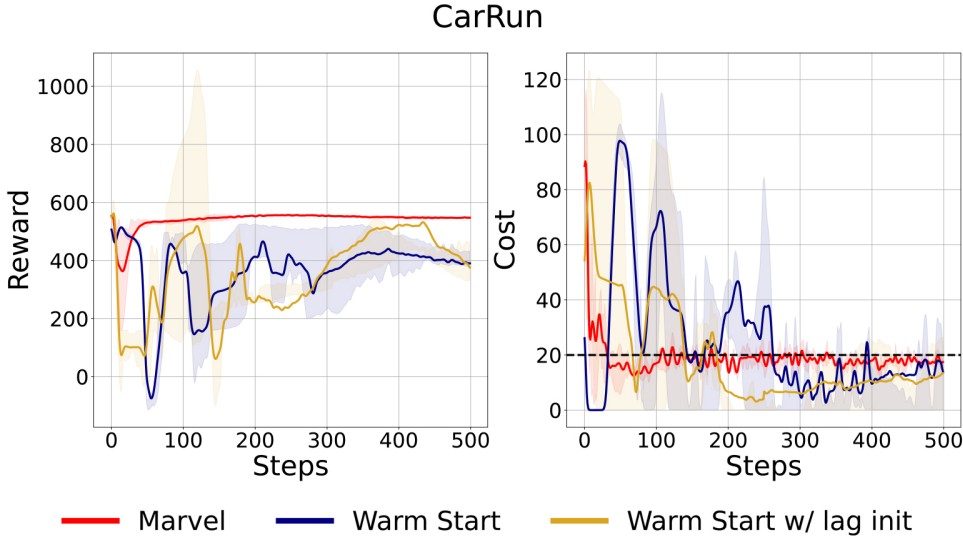

Figure 14: In the figure, "Marvel" and "Warm Start" follow the legend defined in Fig. 1. "Warm Start w/ lag init" represents the approach of empirically selecting appropriate initial values for the Lagrange multipliers and performing online finetuning.

offline data distribution. We then perform off-policy evaluation to distill the Q-values of this cloned policy and use the resulting critics to initialize online finetuning.

However, our experiments show that this approach fails to keep the cost below the safety threshold during online finetuning. Because the BC policy directly imitates actions from the offline dataset, it inevitably learns unsafe action patterns present in the data. When the distilled critics are subsequently used for online updates, these unsafe patterns remain and cannot be corrected even with the adaptive control provided by aPID. As a result, the policy cannot maintain safety and exhibits performance significantly worse than MARVEL.

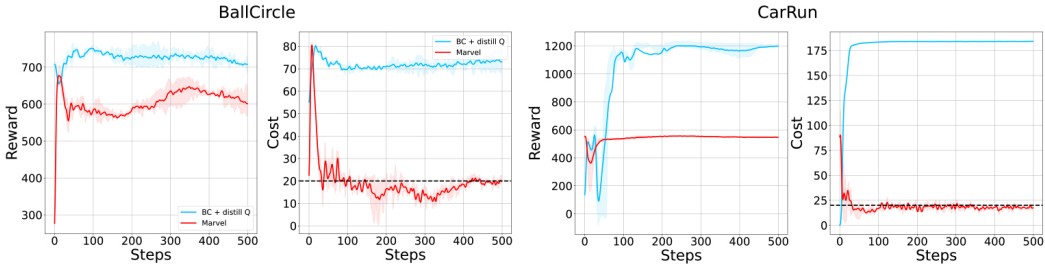

Figure 15: Comparison between Marvel and the additional baseline that distills Q-values of the behavior policy. The BC-distilled baseline fails to control the cost below the threshold due to inherited unsafe patterns from the offline dataset, leading to high constraint violations and lower returns, whereas MARVEL effectively maintains safety and achieves superior performance.

# E More Analysis of Marvel

Similar to the analysis presented in (Liu et al., 2023a), this section introduces an alternative way to evaluate safe RL performance beyond training curves, as shown in Fig. 16. The cumulative cost represents the total cost accumulated from all environment interactions up to a given timestep during training, while the max reward denotes the highest reward achieved up to that timestep. The relationship between these two metrics reflects the algorithm's ability to achieve maximum reward performance under a certain amount of cost incurred in the environment.

The figure shows that Marvel achieves the best max reward for a given cumulative cost. Moreover, when targeting a specific performance level (i.e., reward), Marvel requires the least cumulative cost. This further highlights Marvel's superior performance from another perspective.

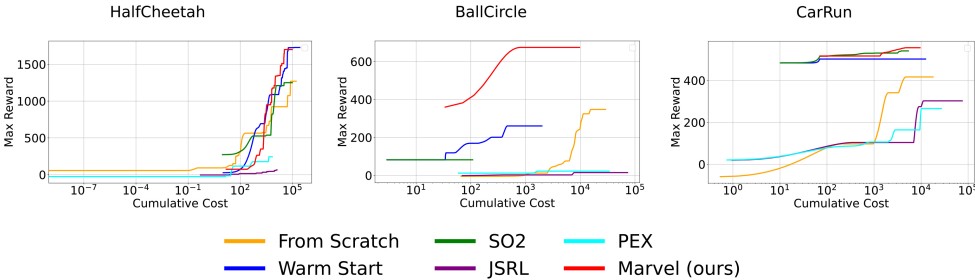

Figure 16: The legends in this figure follow the conventions of this paper and illustrate the relationship between cumulative cost and maximum reward.

# F Additional PID Design Details

For completeness we provide additional details on the PID controller used in Marvel and clarify how each component corresponds to our implementation. We employ a *discrete, position-form* PID update. Let $c_t$ denote the total episode cost at training iteration $t$ and $c_{th}$ the cost limit. The instantaneous violation is

$$e_t = c_t - c_{th}.$$

**EMA-smoothed proportional signal.** To reduce variance, the proportional channel operates on an exponentially smoothed violation:

$$\tilde{e}_t = a_p \tilde{e}_{t-1} + (1 - a_p)e_t,$$

where $a_p \in [0, 1)$ is the smoothing factor.

**EMA-smoothed cost for the derivative channel.** We first compute a smoothed version of the episode cost,

$$\tilde{c}_t = a_d \tilde{c}_{t-1} + (1 - a_d)c_t,$$

with smoothing factor $a_d \in [0, 1)$. This low-pass filtering mitigates the high variance of raw episode costs.

**Lagged derivative approximation.** Instead of estimating the derivative by immediate finite differences, we compute a *lagged* derivative using a delay window of length $d$:

$$D_t = \max\big(0, \ \tilde{c}_t - \tilde{c}_{t-d}\big).$$

This corresponds exactly to our implementation, where we maintain a buffer of the past $d$ smoothed costs and take their difference. Such a delay-based derivative approximation is common in discrete PID controllers, as it substantially reduces sensitivity to noise while retaining responsiveness to sustained increases in cost.

**Anti-windup mechanism.** The integral state is updated as

$$I_t = \max\big(0, \ I_{t-1} + K_i e_t\big),$$

which prevents the integral term from accumulating negative values and ensures stability. This clipping-based anti-windup mechanism is the same as in the implementation.

**Final position-form update.** The Lagrange multiplier used by the policy optimization step is computed as

$$\lambda_t = \max\big(0, \ K_p \tilde{e}_t + I_t + K_d D_t\big).$$

Thus the proportional term uses the smoothed violation $\tilde{e}_t$, the integral term accumulates the raw violation $e_t$, and the derivative term is a lagged difference of smoothed costs. This matches the code path in our implementation and avoids introducing a second integrator.

These design choices (EMA smoothing, lagged derivative, and anti-windup) follow standard practice in discrete PID control and were found to improve robustness and stability in our reinforcement learning setting.

## G Experimental Details

We conduct all experiments in a single Nvidia 4080 GPU.

### G.1 Spearman's rank correlation coefficients

We aim to explore the effect of VPA on the distribution of the Q and Qc networks. Specifically, for different state-action pairs, we need to analyze the true Q and Qc values versus the predicted values from the Q and Qc networks. To achieve this, we choose to use Spearman's rank correlation coefficient, which allows us to quantify the ranking accuracy of the Q and Qc values over a sequence of state-action pairs.

Spearman's rank correlation coefficient, denoted as $\rho$, is a non-parametric measure of the strength and direction of the association between two ranked variables. It evaluates how well the relationship between two variables can be described using a monotonic function, rather than assuming a linear relationship. This makes it particularly useful in our case, where we are more concerned with the rank ordering of predicted versus true Q and Qc values rather than their exact numerical differences.

Mathematically, Spearman's rank correlation coefficient is given by:

$$\rho = 1 - \frac{6 \sum d_i^2}{n(n^2 - 1)} \tag{16}$$

where $d_i$ is the difference between the ranks of the corresponding values of the two variables (in this case, the true and predicted Q or Qc values) for each state-action pair. $n$ is the number of state-action pairs.

Spearman's coefficient ranges from $-1$ to 1, where $\rho = 1$ indicates a perfect positive rank correlation (i.e., the predicted Q and Qc values perfectly match the rank of the true values) $\rho = -1$ indicates a perfect negative rank correlation, $\rho = 0$ indicates no correlation between the ranks of the predicted and true values.

By applying this measure, we can rigorously assess how well the Q and Qc networks preserve the relative rankings of the true values across various state-action pairs, thus quantifying the alignment between the predicted and true distributions.

### G.2 Estimation of Q-values and Qc-values through Monte Carlo Simulations in Table 1

The Q-values represent the expected cumulative reward from a given state when following a specific policy, while the Qc-values represent the expected cumulative cost. These values are estimated through Monte Carlo (MC) simulations, making them accurate because the simulations explicitly capture the sequential interactions of the agent with the environment under the given policy.

For the MC simulations, we use the pre-trained policy derived from the training phase. Each simulation starts from a selected initial state. The number of interaction steps with the environment depends on the specific settings of the environment. For instance, in the BallCircle environment, the maximum number of steps is 200. A total of 10 Monte Carlo simulations are performed, and at each timestep, we record both the reward and the cost. To compute the true Q-values and Qc-values, the recorded rewards and costs are averaged cumulatively across all steps in the episodes.

Regarding the choice of the initial state, the term "dataset" refers to selecting the initial state from the offline dataset used during VPA, whereas "random" indicates that the initial state is chosen randomly. This approach ensures a diverse evaluation and enhances the robustness of the estimated values.

### G.3 Experimental Setup

In Table 6, we present the specific hyper-parameters used in the experiments. Table 7 lists the configurations of the environments used in the experiments.

## H Proof of Theorem 1

We first define a modified VPA Bellman operator with entropy regularization as follows:

$$\mathcal{T}^{\mathrm{VPA}}Q(s,a) = r(s,a) + \gamma \mathbb{E}_{\substack{s' \sim P(\cdot|s,a) \\ a' \sim \pi(\cdot|s')}}[Q(s',a') - \alpha \log \pi(a'|s')] \tag{17}$$

where $\alpha$ denotes the entropy regularization term $\alpha^{\mathrm{VPA}}$ or $\alpha_c^{\mathrm{VPA}}$, depending on the context. We use $\alpha$ for simplicity in following proof. We can have the following result to characterize the contraction property of $\mathcal{T}^{\mathrm{VPA}}$. $\pi_0$ is the policy obtained in the offline phase which is the fixed policy in VPA and also the initial policy of online finetune.

Considering the similarity in the derivations for $Q$ and $Q_c$, we will proceed with the derivation for $Q$, and the derivation for the cost Q-function $Q_c$ follows in the same manner.

**Assumption 1** (Concentrability). $\exists C < \infty$ such that

$$\max_{(s,a) \in S \times A} \frac{d^{\pi_0}(s,a)}{d^{\mu}(s,a)} \leq C$$

where $d^{\pi_0}(s,a)$ denotes the state-action distribution induced by the offline policy $\pi_0$, and $d^{\mu}(s,a)$ is the distribution induced by the behavior policy $\mu$ that generates the offline dataset. For notational simplicity, we refer to the behavior policy's state-action distribution as $\mu$, i.e., $d^{\mu}(s,a) := \mu$.

**Theorem 2** (Contraction Property). The VPA Bellman operator $\mathcal{T}^{VPA}$ is a $\gamma$-contraction under the $L_2$ norm weighted by the state-action distribution $d^{\pi_0}$:

$$\|\mathcal{T}^{VPA}Q_1 - \mathcal{T}^{VPA}Q_2\|_{2,d^{\pi_0}} \leq \gamma\|Q_1 - Q_2\|_{2,d^{\pi_0}}. \tag{18}$$

| Hyper-parameter | Value |
|---|---|
| Policy Learning Rate | 5e-5 |
| Q-network Learning Rate | 3e-5 |
| Qc-network Learning Rate | 8e-5 |
| Lagrangian Learning Rate | 1e-4 |
| SAC-lag: $\alpha$ | 5e-3 |
| VPA Entropy Coefficient : $\alpha$ | 1e-3 |
| VPA Entropy Coefficient : $\alpha_c$ | 5e-4 |
| aPID: Kp | 1e-4 |
| aPID: Ki | 1e-5 |
| aPID: Kd | 1e-5 |
| aPID: $\alpha$ | 0.05 |
| aPID: $\beta$ | 0.05 |
| aPID: $\gamma$ | 0.05 |
| Batch Size | 256 |
| MLP hidden layer size | [256, 256] |
| discount | 0.99 |
| $\tau$ | 5e-2 |
| replay buffer size | 1e6 |

Table 6: Experiment hyper-parameters

| Environment | Episode length | Cost threshold |
|---|---|---|
| BallCircle | 200 | 20 |
| BallRun | 100 | 20 |
| CarCircle | 200 | 20 |
| CarRun | 200 | 20 |
| AntCircle | 500 | 20 |
| AntRun | 200 | 20 |
| DroneCircle | 200 | 20 |
| HalfCheetah | 1000 | 20 |
| Hopper | 1000 | 20 |
| Swimmer | 1000 | 20 |

Table 7: Environment setup

Here, $\gamma \in (0,1)$ is the discount factor, and $\|\cdot\|_{2,d^{\pi_0}}$ is defined as:

$$\|f\|_{2,d^{\pi_0}} := \left( \mathbb{E}_{(s,a)\sim d^{\pi_0}} \left[ f(s,a)^2 \right] \right)^{1/2}.$$

*Proof.* Let $\Delta = Q_1 - Q_2$. Note that the entropy terms cancel when taking the difference between $\mathcal{T}^{\mathrm{VPA}}Q_1$ and $\mathcal{T}^{\mathrm{VPA}}Q_2$, so we have:

$$
\begin{aligned}
&\|\mathcal{T}^{\mathrm{VPA}}Q_1 - \mathcal{T}^{\mathrm{VPA}}Q_2\|_{2,d^{\pi_0}}^2 \\
&= \mathbb{E}_{(s,a)\sim d^\pi}\left[\left(\gamma\,\mathbb{E}_{s'\sim P(\cdot|s,a),\,a'\sim\pi_0(\cdot|s')}[\Delta(s',a')]\right)^2\right] \\
&\leq \gamma^2\,\mathbb{E}_{(s,a)\sim d^{\pi_0}}\left[\mathbb{E}_{s'\sim P(\cdot|s,a),\,a'\sim\pi(\cdot|s')}[\Delta(s',a')^2]\right] \quad \text{(by Jensen's inequality)} \\
&= \gamma^2\,\mathbb{E}_{(s,a)\sim d^{\pi_0}}\left[\mathbb{E}_{(s',a')\sim P(\cdot|s,a)}[\Delta(s',a')^2]\right] \\
&= \mathbb{E}_{(s',a')\sim d^{\pi_0}}[\Delta(s',a')^2] \\
&= \|\Delta\|_{2,d^{\pi_0}}^2.
\end{aligned}
$$

Taking the square root of both sides gives:

$$
\|\mathcal{T}^{\mathrm{VPA}}Q_1 - \mathcal{T}^{\mathrm{VPA}}Q_2\|_{2,d^{\pi_0}} \leq \gamma\|Q_1 - Q_2\|_{2,d^{\pi_0}}.
$$

$\square$

The next result will show that the Q-value function is bounded under the conditions considered in this paper.

**Lemma 1** (Bounded Value Function). *Under the conditions that the maximum reward* $\max|r| \leq 1$ *and* $\max_{(s,a)\in S\times A}|\log\pi_0(a|s)| \leq B$*, the value function of VPA satisfies:*

$$
|Q^{\pi_0}(s,a)| \leq \frac{1+\alpha B}{1-\gamma}. \tag{19}
$$

*Proof.* We start from the entropy-augmented Bellman equation:

$$
Q^{\pi_0}(s,a) = r(s,a) + \gamma\mathbb{E}_{\substack{s'\sim P(\cdot|s,a)\\a'\sim\pi_0(\cdot|s')}}\left[Q^{\pi_0}(s',a') - \alpha\log\pi_0(a'|s')\right].
$$

Taking absolute values and applying triangle inequality:

$$
\begin{aligned}
|Q^{\pi_0}(s,a)| &\leq \mathbb{E}\left[\sum_{t=0}^\infty \gamma^t\left(|r(s_t,a_t)| + \alpha|\log\pi_0(a_t|s_t)|\right)\right] \\
&\leq \sum_{t=0}^\infty \gamma^t(1+\alpha B) \\
&= \frac{1+\alpha B}{1-\gamma}.
\end{aligned}
$$

$\square$

As a direct consequence of the pointwise boundedness of $Q^{\pi_0}$, we can also derive an upper bound on its $L_2$ norm under the state-action distribution $d^{\pi_0}$. Recall that the bound in Lemma 1 is given in terms of the absolute value, i.e., $|Q^{\pi_0}(s,a)| \leq \frac{1+\alpha B}{1-\gamma}$ for all $(s,a) \in S \times A$. Since the $\ell_\infty$-norm is defined as the supremum of the absolute value over the domain, we have

$$
\|Q^{\pi_0}\|_\infty = \sup_{(s,a)}|Q^{\pi_0}(s,a)| \leq \frac{1+\alpha B}{1-\gamma}.
$$

Moreover, for any measurable function $f$ and any distribution $d$, it holds that $\|f\|_{2,d} \leq \|f\|_\infty$. Therefore, we obtain:

$$
\|Q^{\pi_0}\|_{2,d^{\pi_0}} \leq \|Q^{\pi_0}\|_\infty \leq \frac{1+\alpha B}{1-\gamma}. \tag{20}
$$

This result provides a useful uniform bound in the $L_2$ sense, which will be instrumental in the subsequent generalization and convergence analysis.

## H.1 Final Convergence Bound for VPA

we are now ready to quantify how the Q–function produced by $K$ steps of VPA evaluation deviates from the fixed–point solution under the offline policy $\pi_0$.

We begin by briefly reviewing Theorem 1. As the number of iterations $K$ increases, the estimated Q-function $Q_K$ progressively moves away from the initialization $Q_0$, and converges toward the fixed-point solution $\hat{Q}^\pi$ under the offline policy $\pi_0$. The closeness between the two is measured in the weighted $L_2$-norm $\|\cdot\|_{2,d^{\pi_0}}$, which denotes the expected squared error with respect to the state-action distribution induced by policy $\pi_0$.

Based on $\max_{(s,a)\in S\times A} \frac{d^{\pi_0}(s,a)}{d^\mu(s,a)} \le C$ and Lemma 1, after $K$ iterations, with probability at least $1 - \delta$:

$$\|Q_K - \hat{Q}^{\pi_0}\|_{2,d^{\pi_0}} \le \frac{\sqrt{C}\tilde{\epsilon}}{1-\gamma} + \gamma^K \|Q_0 - \hat{Q}^{\pi_0}\|_{2,d^{\pi_0}}. \tag{21}$$

where $\tilde{\epsilon} = \frac{22(1+\alpha B)^2 \log(|\mathcal{F}|/\delta)}{|\mathcal{D}|} + 20 d_F^{\pi_0,\text{VPA}}$. $\delta \in (0,1)$ is the confidence level parameter, $\mathcal{F}$ denotes the function class used to approximate Q-values, and $|\mathcal{D}|$ is the number of samples in the dataset. The term $d_F^{\pi_0,\text{VPA}}$ represents the inherent Bellman evaluation error under policy $\pi_0$ using the VPA operator. The parameter $\alpha$ corresponds to the entropy coefficient in the VPA Bellman operator, and $B = \max_{(s,a)} |\log \pi_0(a|s)|$ denotes the maximum policy entropy. The constant $C$ comes from the concentrability assumption, and $\gamma \in (0,1)$ is the standard discount factor in Markov decision processes.

*Proof.* We decompose the error at iteration $k$ using triangle inequality:

$$\|Q_k - \hat{Q}^{\pi_0}\|_{2,d^{\pi_0}} \le \underbrace{\|Q_k - \mathcal{T}^{\text{VPA}} Q_{k-1}\|_{2,d^{\pi_0}}}_{\text{(I) Approximation error}} + \underbrace{\|\mathcal{T}^{\text{VPA}} Q_{k-1} - \hat{Q}^{\pi_0}\|_{2,d^{\pi_0}}}_{\text{(II) Iteration error}}.$$

**Term (I):** Based on the concentrability,

$$\text{(I)} \le \sqrt{C}\|Q_k - \mathcal{T}^{\text{VPA}} Q_{k-1}\|_{2,\mu}.$$

At each iteration $k$, the VPA algorithm performs a supervised regression to update the Q-function estimate $Q_k$, using regression targets constructed from the previous iterate $Q_{k-1}$ under an VPA Bellman operator. Given the offline dataset $\mathcal{D}$, the Q-function is updated one step to minimize the following objective:

$$Q_k \leftarrow \arg\min_{Q\in\mathcal{F}} \sum_{i=1}^{|\mathcal{D}|} [Q(s_i, a_i) - y_i]^2$$

where $\mathcal{F}$ denotes the function class used to approximate Q-functions, and the target value $y_i$ incorporates an entropy penalty defined as:

$$y_i = r_i + \gamma \left( \mathbb{E}_{a'\sim\pi_0(\cdot|s_i')} [Q_{k-1}(s_i', a') - \alpha \log \pi_0(a'|s_i')] \right).$$

The training objective seeks to minimize the squared deviation between the predicted Q-values and the soft Bellman target computed from $Q_{k-1}$ at each iteration.

To bound $|y_i|$, using $\|Q_{k-1}\|_\infty \le \frac{1+\alpha B}{1-\gamma}$ (see Lemma 1) and the boundedness assumptions $|r_i| \le 1$, $\max_{(s,a)\in S\times A} |\log \pi_0(a|s)| \le B$, we obtain:

$$|y_i| \le |r_i| + \gamma (\|Q_{k-1}\|_\infty + \alpha B)$$
$$\le 1 + \gamma \left( \frac{1+\alpha B}{1-\gamma} + \alpha B \right)$$
$$\le 2 \cdot \frac{1+\alpha B}{1-\gamma} = 2\bar{V}.$$

To control the one-step regression error, we apply the least squares generalization bound (Lemma 11) from (Agarwal et al., 2019), which provides a high-probability upper bound on the expected squared error between the regression output $Q_k$ and the target $y_i$ computed from $Q_{k-1}$.

Specifically, given that (1) the supervised regression target $y_i$ is uniformly bounded, i.e., $|y_i| \leq 2\bar{V}$ (2) the function class $\mathcal{F}$ used to fit $Q_k$ is finite or has bounded covering number (3) the sample inputs $(s_i, a_i)$ are drawn i.i.d. from the offline dataset, we can get the following bound with probability at least $1 - \delta$:

$$\|Q_k - \mathcal{T}^{\mathrm{VPA}} Q_{k-1}\|_{2,\mu}^2 \leq \frac{22\bar{V}^2 \log(|\mathcal{F}|/\delta)}{|\mathcal{D}|} + 20 d_F^{\pi_0, \mathrm{VPA}}$$

where $d_F^{\pi_0, \mathrm{VPA}}$ denotes the approximation error between the best function in $\mathcal{F}$ and the Bellman target under policy $\pi_0$.

Substituting $\bar{V} = \frac{1 + \alpha B}{1 - \gamma}$ into the bound above:

$$\|Q_k - \mathcal{T}^{\mathrm{VPA}} Q_{k-1}\|_{2,\mu} \leq \sqrt{\frac{22(1 + \alpha B)^2 \log(|\mathcal{F}|/\delta)}{|\mathcal{D}|(1 - \gamma)^2} + 20 d_F^{\pi_0, \mathrm{VPA}}} = \sqrt{\tilde{\epsilon}}.$$

Thus, we can get the bound of approximation error (I):

$$\text{(I)} \leq \sqrt{C} \|Q_k - \mathcal{T}^{\mathrm{VPA}} Q_{k-1}\|_{2,\mu} \leq \sqrt{C\tilde{\epsilon}}.$$

**Term (II):** Based on Theorem 2, we have:

$$\|\mathcal{T}^{\mathrm{VPA}} Q_{k-1} - \hat{Q}^{\pi_0}\|_{2,d^{\pi_0}} = \|\mathcal{T}^{\mathrm{VPA}} Q_{k-1} - \mathcal{T}^{\mathrm{VPA}} \hat{Q}^{\pi_0}\|_{2,d^{\pi_0}} \leq \gamma \|Q_{k-1} - \hat{Q}^{\pi_0}\|_{2,d^{\pi_0}}.$$

**Combining both (I) and (II) term:**

$$\|Q_k - \hat{Q}^{\pi_0}\|_{2,d^{\pi_0}} \leq \sqrt{C\tilde{\epsilon}} + \gamma \|Q_{k-1} - \hat{Q}^{\pi_0}\|_{2,d^{\pi_0}}.$$

Let $Q_0$ be the Q-function learned offline, which is the starting point for VPA. Unroll the recursion:

$$\begin{aligned}
\|Q_K - \hat{Q}^{\pi_0}\|_{2,d^{\pi_0}} &\leq \sqrt{C\tilde{\epsilon}} + \gamma \|Q_{K-1} - \hat{Q}^{\pi_0}\|_{2,d^{\pi_0}} \\
&\leq \sqrt{C\tilde{\epsilon}} + \gamma \left( \sqrt{C\tilde{\epsilon}} + \gamma \|Q_{K-2} - \hat{Q}^{\pi_0}\|_{2,d^{\pi_0}} \right) \\
&= \sqrt{C\tilde{\epsilon}}(1 + \gamma) + \gamma^2 \|Q_{K-2} - \hat{Q}^{\pi_0}\|_{2,d^{\pi_0}} \\
&\leq \sqrt{C\tilde{\epsilon}}(1 + \gamma + \gamma^2) + \gamma^3 \|Q_{K-3} - \hat{Q}^{\pi_0}\|_{2,d^{\pi_0}} \\
&\vdots \\
&\leq \sqrt{C\tilde{\epsilon}} \sum_{k=0}^{K-1} \gamma^k + \gamma^K \|Q_0 - \hat{Q}^{\pi_0}\|_{2,d^{\pi_0}} \\
&= \frac{\sqrt{C\tilde{\epsilon}}}{1 - \gamma}(1 - \gamma^K) + \gamma^K \|Q_0 - \hat{Q}^{\pi_0}\|_{2,d^{\pi_0}} \\
&\leq \frac{\sqrt{C\tilde{\epsilon}}}{1 - \gamma} + \gamma^K \|Q_0 - \hat{Q}^{\pi_0}\|_{2,d^{\pi_0}}.
\end{aligned}$$

$\square$

# I Limitation

While Marvel performs well in most environments, it does not exhibit the same effectiveness in certain scenarios, such as in the AntRun environment. As depicted in Fig. 5, during finetuning, Marvel does not significantly improve cost and reward metrics. Consequently, aPID evidently does not function optimally in these settings. This suggests that further enhancements are needed for VPA to increase the agent's exploratory behavior during online finetuning. Combined with aPID's efficient control over costs, this approach could achieve optimal performance with minimal interaction with the environment, thus minimizing the time required. In addition, it would be interesting to investigate strategies that explicitly re-initialize or reset $K_d$ when the environment undergoes new phases of non-stationarity.

