# OpenReview forum: "Towards Fast Safe Online Reinforcement Learning via Policy Finetuning"
_TMLR — Accepted by TMLR_

### Review · Reviewer_nZsR · 2025-10-21

**Summary Of Contributions:**

The paper proposes an effective method for offline-to-online finetuning in the context of safe RL, where in addition to maximizing a reward, an agent needs to stay within a certain cost budget, which measures an important constraint on the policy. The authors identify core challenges in handling the offline-to-online transfer, pessimistic values, and wrong Lagrangians, and introduce methods to address these.

**Audience:**

Yes

**Audience Explanation:**

Overall, the paper seems reasonable and (mostly) complete to me. The discussed issues seem of general relevance to the (sub)community, and the solutions fit the issues.

**Claims And Evidence:**

Yes

**Claims Explanation:**

Overall, the paper is well written and easy to follow. Some of the writing is a bit dense, for example I would encourage the authors to use enumerate as they frequently use numbered list which are inlined in text. This is a small issue though. The acronym of the method also feels extremely forced and is effectively unrelated to the name of the method. While the authors are free to pick any name of course, something a bit more aligned with the actual title might make it more memorable and easy for others to look for.

**Requested Changes:**

The problem discussion that motivate the warm start method is reminiscent of Zhou et al. (https://openreview.net/forum?id=HN0CYZbAPw), even though the solution is different. I actually somewhat prefer the solution suggested here. I suggest the authors briefly acknowledge and discuss the related work. It would also be interesting if retaining the offline buffer helps in some sense in the safe RL setting, which would justify keeping it around even better.

The PID controller introduction seems somewhat orthogonal to the offline-to-online setting. While I appreciate the difficulty of estimating the Lagrange multipliers from offline data, the PID method would probably work in both the offline-to-online and the pure online setting. I think it would be great if the authors experimented with the PID method as well in the standard pure online setting, as this might be of independent interest to researchers focused on that area.

---

> ### Author Response · Authors · 2025-11-25
> **Response to Reviewer nZsR**
>
> We would like to thank Reviewer nZsR for the valuable suggestions on related work,
> the role of the offline buffer, and the connection between PID control and pure online safe RL.
> Your feedback led us to expand the Related Work section, highlight the baseline **SAC with an offline replay buffer**,
> and clarify how aPID builds upon PID-based Lagrangian updates in online settings.
>
> ---
>
> # **Q1: discuss of related work**
>
> **A:** Thank you for the helpful suggestion. We have added the cited paper [1] to the **Related Work** section and briefly discussed how it works.
>
> # **Q2: if retaining the offline buffer helps online finetune**
>
> **A:** Thank you for the insightful comment. To examine whether retaining the offline buffer benefits the offline-to-online process, we have included a comparison in Appendix (Fig. 6) between Marvel and the baseline **SAC with an offline replay buffer**. The results show that reusing the offline buffer consistently yields a policy whose cost remains above the threshold, likely because the distribution of the offline dataset differs from that of the online interactions. This further confirms the advantage of our proposed O2O training framework.
>
>
> # **Q3: PID control in pure online safe RL**
>
> **A:** Thank you for the valuable suggestion. As shown in [2], PID-based Lagrangian methods have proven to be a promising and effective approach for controlling the dual variable in pure online safe RL.
>
> # **Q4: aPID appears orthogonal to the offline-to-online setting.**
>
> **A:** aPID stabilizes Lagrange updates in general and can be beneficial in pure online safe RL.
> In O2O it is particularly valuable because VPA relaxes offline conservatism and increases early exploration of high-return state–action pairs, which raises violation risk during early finetuning.
> The gain schedule supplies strong early correction and lower late-stage gains for damping, thereby shortening the transition from overshoot to stable operation. Therefore, the design of aPID is closely integrated with VPA, and their joint effect is what enables more efficient online finetuning.
>
> [1] Zhou, Zhiyuan, et al. "Efficient online reinforcement learning fine-tuning need not retain offline data." arXiv preprint arXiv:2412.07762 (2024).
>
> [2] Stooke, Adam, Joshua Achiam, and Pieter Abbeel. "Responsive safety in reinforcement learning by pid lagrangian methods." International Conference on Machine Learning. PMLR, 2020.

---

### Review · Reviewer_rko7 · 2025-10-26

**Summary Of Contributions:**

The paper addresses the problem setting of offline-to-online safe RL, which requires trading off between reward maximization and constraint satisfaction in a way that most efficiently makes use of both online samples and a pre-existing offline dataset. The paper identifies two potential challenges in applying existing methods to the offline-to-online problem: 1) inaccurate Q-value and cost predictions, and 2) inappropriately initialized Lagrange multipliers for the online fine-tuning period. It proposes a method to mitigate both of these issues by applying a policy evaluation phase during offline learning and using a PID controller to adapt the lagrange multipliers during the online phase.

**Strengths**

- The paper successfully isolates two independent challenges in adapting existing methods to the offline-to-online setting and conducts ablations to show that both of these issues must be addressed to obtain optimal performance.

- The proposed PID controller for the Lagrange multiplier is a creative idea for efficiently recovering from inappropriate multiplier values.

- Learning a policy from offline data that is efficient to fine-tune is a challenge that is relatively under-studied in RL, and this paper takes an interesting perspective on the challenge by also considering the additional complexity of added cost constraints.

- The proposed method is relatively simple (aside from the number of additional hyperparameters it introduces, more on that later) and the paper provides a clear story to explain why it works.

**Weaknesses**

- The introduction poses offline to online RL as a "pretraining to finetuning" paradigm, but it is not exactly comparable to the traditional pretraining setting which uses a much larger and more diverse dataset for pretraining and then finetunes on a subset of that dataset.

- The mismatch between the objective of offline safe RL vs offline-to-online safe RL suggests that offline safe RL methods should trivially be worse than any method which assumes access to a fine-tuning period. This makes the comparison between the proposed approach and offline RL methods feel unfair.

- Value mismatch is a natural consequence of these different objectives; offline RL methods are deliberately pessimistic about the values of off-policy actions to reduce the risk of overestimation / bootstrapping errors. Inaccurate Q-estimation is a feature of such methods, not a bug! When using the offline phase as a 'warm-up', it is obvious that one does not need to be as conservative about off-policy action values and costs. Additionally, how conservative one wants to be will now depend on the fine-tuning budget allotted to the learner.

- The proposed 'value alignment' phase is just on-policy value estimation of the data collecting policy. It seems like what the paper's results are saying is just that if you're going to fine-tune afterwards, you don't need to do any fancy offline RL methods and can just estimate the Q-values.

- The Lagrange multiplier adaptation method introduces by my count at least 9 additional hyperparameters (alpha/beta/gamma and kmin/max for each K), which makes me question how robust the observed gains are. I'm not convinced by the empirical evaluations on two environments that the aPID is improving performance at a rate that is "worth" the added complexity.

- The BallCircle comparison in Figure 1 just seems like none of the methods are actually learning (warm start gets less reward than MARVEL off the bat).

- In Figure 3, the PID controller obtains similar costs as the aPID controller, but worse rewards, which seems odd to me as I would have expected the Lagrange multiplier to have roughly symmetric effects on reward and cost.

- Empirical evaluations show that the adapted baseline methods dramatically underperform the offline learning baseline in almost every environment. This is surprising to me as I would have assumed that the O2O methods were explicitly designed to outperform an offline baseline, and makes me suspect that these methods were not given a fair shot in the evaluations (perhaps due to insufficient hyperparameter tuning.

- I would have liked a clearer sense of how good the data collection policy that generated the offline examples was, and how much the benefits of MARVEL over from scratch / warm start / offline depend on the offline data collection policy. I would assume that the value alignment phase depends heavily on the original policy being reasonably good.

**Additional Comments:**

One follow-up question I have about the value alignment phase is whether applying an offline RL method prior to the policy evaluation objective used for value function alignment even matters. If it does matter, does that mean that the policy evaluation phase needs to be limited in duration to avoid overwriting too much of the conservatism that was baked in originally? This is an important ablation that I would want to see in the paper, ideally demonstrating the sensitivity of the final performance to the length of the value alignment phase.

**Audience:**

Yes

**Audience Explanation:**

In general, I think many people are interested in understanding how best to leverage offline data to facilitate sample efficient online learning. Ensuring that the learner also satisfies a cost constraint is similarly desirable in several problem settings.

**Claims And Evidence:**

No

**Claims Explanation:**

The paper does not provide sufficiently convincing and clear evidence to support its conclusions in its current form. In particular:

1. It's not clear how one should budget the offline RL pretraining and value alignment phases, or why policy evaluation is such a better starting point for online fine-tuning than offline methods. This seems like something that should have already turned up in the literature if it were a robust result.

2. The number of hyperparameters involved in the aPID controller make me suspicious of the relatively small gains it appears to produce in the experimental results section.

3. In general, the number of environments used, particularly for ablations, is fairly small and non-diverse, increasing the risk of the results being an inadvertent artifact of differentially motivated tuning of hyperparameters.

**Requested Changes:**

The paper requires two major changes before I think it can be said to have rigorously shown that the proposed method is meaningfully better than a naive combination of existing approaches.

1. The study of value alignment should demonstrate a clearer trade-off between effectiveness of the purely offline policy (which will be more conservative) and the value pre-aligned policy (which may be more vulnerable to going out of distribution but will also be a more accurate estimate of the true Q-value). One additional useful baseline would be to simply distill on the q-values of the policy which generated the data and see whether training on that from scratch does better than the slightly less direct value alignment phase which also aims to accurately estimate the q-values of the current policy. The benefits of this approach should also be stress-tested against poorly-performing initial policies in deterministic environments, where a good offline learning algorithm should be able to stitch together trajectories and learn a better initial policy than what would be obtained by policy evaluation.

2. The benefits of the aPID controller need to be decisively shown to outperform what would be expected from tuning the additional degrees of freedom in the hyperparameter set.

---

> ### Author Response · Authors · 2025-11-25
> **Response to Reviewer rko7**
>
> We sincerely thank Reviewer rko7 for the thorough and insightful feedback. Your comments helped us clarify the motivation of the offline-to-online safe RL setting, refine the explanations of VPA and aPID, and improve the presentation of figures and experimental details. We have carefully addressed all points and revised the manuscript accordingly.
>
> ---
>
> # Requested Changes
>
> **A:**
> **'Value pre-aligned policy' vs. 'purely offline policy':**
> We realize that our description may have been unclear. To clarify, VPA does **not** change the policy; it recalibrates the reward/cost critics on the offline support. Thus the 'pre-aligned policy' and the 'purely offline policy' are the same actor $\\pi_0$. The O2O pipeline is: (i) train an offline safe RL policy and critics, (ii) apply VPA to the critics (policy unchanged), (iii) perform online finetuning with a primal–dual safe RL algorithm (we instantiate SAC-lag).
>
> **Additional baseline that distill q-values of the policy which generated the dataset:**
> We would like to clarify that in offline RL, the dataset is not always generated by a single policy; it can be a mixture of trajectories collected from multiple behavior policies. We test on BallCircle and CarRun, two deterministic environments(actually all environments we use in this paper are deternimistic). As shown in **Appendix D.7**, to align with this mixed dataset, we first use behavior cloning to obtain a policy consistent with the offline data distribution. We then perform off-policy evaluation to distill the Q-values of this cloned policy. However, we found that this approach fails to satisfy the cost threshold, even when employing aPID as a strong mechanism for adjusting the Lagrange multiplier.
>
> # **Q1: Clarification of the 'pretraining to finetuning' phrasing**
>
> **A:** Thank you for the insightful point. Our intent was to highlight that, in the O2O safe RL setting, real-world interaction can be expensive or risky. We therefore rely on a **fixed, pre-collected** offline dataset to learn a strong warm start and then use a **small** online interaction budget to finetune under constraints. We agree this differs from classical NLP/CV pretraining on massive, diverse corpora. In the revision, we will adjust the Introduction to explicitly distinguish O2O from large-scale pretraining and to frame offline training as a **warm-start** step tailored to an online-constrained objective.
>
> # **Q2: Clarification of including pure offline safe RL as a baseline**
> **A:** We include the offline-only result because it is exactly the starting point for O2O finetuning, and in practice online updates can occasionally reduce performance (e.g., instability or forgetting). Reporting the offline score establishes a clear reference under the same evaluation protocol and makes it transparent whether O2O truly improves upon the warm start.
>
> # **Q3: Clarification on value mismatch and the role of conservatism**
> **A:** We agree that pessimism is a **feature** for pure offline RL: it controls extrapolation error and often yields a reliable policy on the dataset support. Furthermore, without conservatism, the offline phase would fail to learn a good policy, and consequently, it would not provide a meaningful warm start for online finetuning. Our VPA aligns the critics on the offline support, moderately optimistic for reward, conservative yet accurate for cost, so that the critics are better matched to the subsequent Lagrangian finetuning. This mitigates warm-start mismatch while retaining the benefits of offline conservatism.
>
> # **Q4: BallCircle in Figure 1**
> **A:** We appreciate the concern. We used five random seeds in our experiments. The poor performance of the warm start precisely motivated our algorithm design. The seemingly lower starting point of the warm start curve occurs because we applied the same smoothing to all training curves; since Marvel’s curve rises much faster, the smoothed initial point appears lower. In fact, they start from the same value. Please refer to Table 2 for the actual performance data.
>
> # **Q5: On the number of aPID hyperparameters**
> **A:** Indeed, we introduced additional parameters, but the settings for $K_{\\min}$ and $K_{\\max}$ are fixed to default, 0.1× and 10× the initial value. Moreover, the hyperparameters $\\alpha$, $\\beta$, and $\\gamma$ are kept consistent across all environments, which demonstrates the robustness of Marvel. Figure 11 also shows that selecting these three parameters is straightforward.

---

> ### Author Response · Authors · 2025-11-25
> **Response to Reviewer rko7**
>
> # **Q6: aPID vs. PID in Figure 3**
> **A:** We believe this phenomenon illustrates the advantage of aPID over PID. aPID can adjust $\\lambda$ more precisely. Since the policy update depends on $\\lambda$ to balance reward and cost, $\\lambda$ affects every timestep. aPID achieves better performance precisely because it provides a more effective mechanism for adjusting $\lambda$. For experimental completeness, we additionally included two environments, **CarRun and HalfCheetah**. The supplementary experiments further confirm the soundness and effectiveness of our algorithmic design.
>
> # **Q7: Performance of baselines**
>
> **A:** We used identical seeds, cost threshold and evaluation protocols across methods; ranges followed the original papers. Since offline-to-online safe RL is still an unexplored area, our baselines are selected from offline-to-online RL algorithms that do not specifically address safety constraints. As a result, they cannot achieve better performance. Nevertheless, some adapted O2O baselines do perform competitively on certain tasks (e.g., PEX on AntRun, SO2 on DroneCircle). Our aim is improved **transients and final performance**, which Marvel achieves.
>
> # **Q8: Marvel under different offline policy qualities**
>
> **A:** We agree that dataset quality matters. Figure 9 varies the quality by mixing different ratios of random-policy data in two environments. Across these mixtures, Marvel remains robust and consistently outperforms from-scratch and other baselines.

---

### Review · Reviewer_5PFa · 2025-11-13

**Summary Of Contributions:**

The paper proposes offline-to-online (O2O) "safe" RL with Value Pre-Alignment to initialise the Q-functions ($Q$ and $Q_c$) w.r.t. the learned offline policy and use them with adaptive PID (aPID) to effectively adjust and learn the Lagrange multiplier $\lambda$. The contribution comes with theoretical evidence (e.g. Theorem 1) and a large number of experimental supports, but my major concern is that the PID control part (and thus its adaptation rules) are incorrectly derived, although I value the authors' ideas and efforts.

**Audience:**

Yes

**Audience Explanation:**

This work particularly tackles the difficulty in O2O safe RL, which embraces the topics of RL such as offline RL, safe RL and offline and online transfer. The PID technique is quite control-engineering-oriented but has already been also studied in ML community (Stooke et al., 2020) in relevant works on safe RL. So, TMLR's audience, particularly in the fields of RL, shall be interested in the findings of the paper if the authors have corrected the error in their PID control explained below.

**Broader Impact Concerns:**

The paper deals with "safe" RL, which is applicable to safety-critical domains, but safety is not guaranteed during the learning phase. The offline RL may compensate this, but during the exploratory online phase, there are certain cases and chances of inevitably violating the safety margin as it is evidenced by the experimental results. So, a special care is needed when the proposed method is applied to safety-critical domains as the safety is still not guaranteed during online learning, especially at the early stage.

**Claims And Evidence:**

No

**Claims Explanation:**

The authors have demonstrated the performance of the proposed approach with a large number of experiments including ablation studies, correctness checks and sensitivity analysis. The bound of the VPA Q-functions from the fixed points are given as a Theorem whose proof seems reasonable. However, as mentioned above, the PID control part (and thus its adaptation rules) are incorrectly derived. Their PID control (12) is *not* a PID control but a variant, so there is a lack of evidence that supports the design of the PID adaptation and their PID control part itself.

**Requested Changes:**

**[critical/main concerns on the PID control and its adaptation rules]**

1. First of all, PID control (12) is *not* a PID control but a variant. Specifically, denoting the error $e_t := C(\pi_t) - c_{th}$, we can view the Lagrangian update (3), $\lambda_{t+\Delta t} = \lambda_t + \alpha_\lambda e_t$, as an Euler-discretisation of the Integral control $\lambda_t = k_i \int_0^t e_\tau d\tau$, with $\alpha_\lambda = \Delta t \cdot k_i$, as the time derivative satisfies $(\lambda_{t+\Delta t} - \lambda_t) /\Delta t \approx \dot \lambda_{\Delta t + t}$ $= k_i e_{t+\Delta t}$ for small $\Delta t > 0$. That is, $K_p$ in (12) must be the integral gain $k_i$ and there is no motivation of using $K_i$ in (12) which is actually a "double" integral gain that is not a part of PID control and rarely introduced to controllers for stability reason. In fact, the authors should've not started their derivation from (12) which is a mixture of discrete and continuous time. What Stooke et al. (2020) in the reference showed in their work is to propose the P and D gains attached to the Integral part "$k_i \int_0^t e_\tau d\tau$" (Secs 4.2 and 4.3), so that the PID-control-oriented Lagrange update (when discretised) must have $k_p$, $k_i$ and $k_d$ terms associated with $e_t$ and its first- and second-order differences $\Delta e_t$ and $\Delta^2 e_t$, respectively. *In summary, the derivation needs to be corrected, and the authors have not introduced the $k_d$-part associated with $\Delta^2 e_t$ but the seemingly unnecessary double integrator (the $K_i$ part); the introduction of the double integrator has no motivation, so I believe that $K_i$-term contributes no significant effect on the learning performance.*

2. Since the PID part has not been designed properly, neither has "adaptive" PID (aPID) been, although aPID dramatically improves the learning performance. Specifically, (1) the design of the $K_i$-adaptation (14) does not make sense; (2) the $K_p$-adaptation (13) in fact corresponds to the adaptation rule of the integral term $k_i$ that is however introduced to eliminate the steady state (i.e., final) error, rather than speeding up the convergence, so the adaptation rule (13) is an unprincipled heuristic; (3) the $K_d$-rule (15) corresponds to $k_p$-adaptation rule, hence it is reasonable to design it in the way (13) and (14) are designed, but it has not done. Rather, $K_d$-rule (15) depends on the std dev $\sigma_c$ (proportionally) and the average cost $\bar c$ (inv. proportionally); it is not adaptive in the sense that it is always increasing as both $\sigma_c$ and $\bar c$ are always positive. I suspect that the $K_d$ gain will always increase from its minimum $K_{d_{min}}$ to maximum $K_{d_{max}}$, probably very rapidly at the early stage. Have the authors checked the trajectories of $K_d$?

Because of those two critical concerns, I lean to suggest a "reject", although I appreciate a numerous number of experiments to verify the performance, unless there is a significant improvement/revision including the derivations of the PID control and its adaptation rule and the experiments corresponding to PID/aPID control. Other less significant concerns regarding PID control are as follows.

3. I believe the clipping thresholds (bounds) of $K_p$, $K_i$ and $K_d$ are critical. The authors mentioned that the hyper-params $\alpha$, $\beta$ and $\gamma$ are easy to tune and the same across all environments (as shown in the Appendix), but how did authors tune the bounding parameters (e.g. $K_{p_{min}}$ and $K_{d_{max}}$). Are they also the same across the environments?

4. A limitation of the PID approach in principle is that the sampling time $\Delta t$ is sufficiently small, but there are many other tasks where $\Delta t$ is not so or is even not defined (e.g. the game of Go). The authors are encouraged to mention this somewhere, e.g., Conclusions or Limitations in Appendix H.

**[Minor Concerns/Corrections on Theorem 1]**

1. $\delta$ in the phrase "with probability at least $1 - \delta$" affects the approximation error $\tilde \epsilon$ according to the proof but has not mentioned in Theorem 1. What is the implication of that dependancy? It'd be great if the authors mention it or add some explanation about $\delta$ within Theorem 1 or the next paragraph.

2. On pp. 5, "Theorem 1 further implies that a larger $\alpha^{VPA}$ should be selected in Eq. 8 for reward Q-values, whereas $\alpha^{VPA}_c$ in Eq. 9 should be smaller to ensure a more accurate estimation of cost Q-values …" — why? It may need some additional explanations… What if $\alpha^{VPA}_c$ is large? Does the behaviour become more conservative?

3. In Theorem 1, "$\epsilon$ and $\tilde \epsilon_c$ are the approximation errors that decrease the size of the dataset while increasing with the values of $\alpha^{VPA}$ and $\alpha^{VPA}_c$, respectively." — some errata in English? “decrease with the size of the dataset”?

**[Other Minor Concerns]**

1. It seems obvious that there is relevancy between $L_{Q_c}^{CPO}$ on pp. 4 and (6) and (7), among the $\psi$-terms, especially between $\mathcal{P}(\cdot)$ in (7) and $Q_c(\cdot)$ in $L_{Q_c}^{CPO}$. Can you explain it to me and if possible, briefly in the manuscript?

2. On pp. 5, "The sparsity of oﬄine cost leads to a pretrained cost Q-function that predicts low costs for IND state-actions, further limiting exploration during finetuning." — I don’t understand this… if the costs are low, then how it is limiting explorations? In my opinion, low costs increases possibility of constraints satisfaction, so more chance of exploring the region as the agent recognises those regions as being safe.

**[Other Minor Corrections]**

1. The authors have used different notations to represent the input distributions, namely, in Sec. 2, $s \sim d$ and $(s, a, s') \sim d$ in Online Safe RL subsection, and $D$ in Offline Safe RL part; both $D$ and $\mathcal{D}$ in Sec. 3. There would have been some intention of using those different notations, but they seem too much and somewhat unnecessary. I suggest to use the same and/or consistent notations. E.g. right below (9), the authors have mentioned that $(s, a, s') \sim D$ for offline data; it'd be good to explain when the notation $D$ appears for the first time in Sec. 2; I don't understand why $\mathcal{D}$ is introduced in (6) and (7) rather than $D$ is used.

2. Right below (15), what is the cost $c_i$ exactly? $Q_c(s, a)$?

3. What is $\mathcal{L}_2$ in Eqs. (8) and (9)? If it meant an L-2 norm, why don’t we employ the standard notation $||\cdot||_2$; otherwise, we need a definition of it at least.

4. In Algorithm 1 (Appendix A), what is the final policy? Just argmax of the Q? For clarity, it’d be great to mention the final policy within the algorithm, although it is mentioned in Sec. 4.1. The same for offline RL loop.

5. pp. 10: "To further verify this, the last two columns of table 2 show …" —  change “columns” to “rows”

---

> ### Author Response · Authors · 2025-11-25
> **Response to Reviewer 5PFa**
>
> We deeply appreciate Reviewer 5PFa’s detailed technical feedback and constructive suggestions on the PID formulation and theoretical analysis. Your comments prompted us to correct ambiguous notation, provide discrete position-form PID derivations,
> and include additional explanations and sensitivity analyses for aPID. These revisions have made the paper both clearer and more rigorous.

---

> ### Author Response · Authors · 2025-11-25
> **Response to Reviewer 5PFa**
>
> # critical/main concerns on the PID control and its adaptation rules [part 1]: Clarification on Eq. (12) and the ``double integrator'' concern.
>
> **A:** We acknowledge that the notation of our original Eq. (12) was misleading. Our implementation itself is correct, as shown in the code from supplementary material, which is modified from [3] and exactly match [1]; however, it was described inaccurately in the text.
> It mixed a discrete dual-ascent update ($\lambda_{t+1}=\lambda_t+\cdots$) with continuous-time terms ($\int e(\tau)d\tau$ and $\tfrac{de(t)}{dt}$),
> which incorrectly suggested that $\lambda_t$ itself integrates the error once more in addition to the integral term.
> This notation indeed makes it appear as a ``double integrator'', although that was never intended.
>
> In the revision, we have corrected this formulation.
> Our implementation actually follows the \emph{discrete, position-form PID-Lagrangian} proposed in [1]:
> $$
> I_t = \\max\(0, I_{t-1} + K_i\, e_t), \\
> D_t = \\max\(0, \tilde c_t - \tilde c_{t-d}).
> $$
> The multiplier is then
> $$
> \\lambda_t = \\max (0, K_p \\tilde e_t + I_t + K_d D_t).
> $$
> This structure contains a single integral channel (through $I_t$) and therefore no double integration. It exactly matches the position-form PID used in prior work. In the implementation we further include standard stabilizers, which is EMA smoothing of signals, derivative lag, anti-windup clipping, and bounded gains, which do not alter the control law but improve robustness in stochastic RL environments.
>
> We appreciate the reviewer’s observation and we have corrected the equation, clarified the discrete form.
>
> # critical/main concerns on the PID control and its adaptation rules [part 2]: Clarification on the design rationale of the aPID gain scheduling.
>
> **A:** We thank the reviewer for the detailed feedback on the adaptation rules for $(K_p, K_i, K_d)$.
> Our design is heuristic rather than derived from a strict control-theoretic adaptation law, but it follows standard *gain-scheduling* principles in digital PID controllers and has shown strong empirical robustness in the O2O safe RL setting.
>
> **1. $K_p$ and $K_i$**
> The updates for $K_p$ and $K_i$ adjust the controller’s low-frequency gain according to the magnitude of constraint violation:
> $$
> K_p \\leftarrow \\mathrm{clip}\\Big(K_p \\cdot (1 + \\alpha \\tfrac{\\bar c - c_{th}}{\\bar c}),\\, K_{p,\\min}, K_{p,\\max}\\Big), \quad
> K_i \\leftarrow \\mathrm{clip}\\Big(K_i \\cdot (1 + \\beta \\tfrac{\\bar c - c_{th}}{\\bar c}),\\, K_{i,\\min}, K_{i,\\max}\\Big).
> $$
> Here $\\bar c$ denotes the average cost over the most recent $n$ samples stored in a fixed-length `deque`. When $\\bar c > c_{th}$, both gains increase to accelerate correction of violations; when $\\bar c < c_{th}$, they decrease slightly to prevent overshoot. This heuristic captures the intended "early strong–later gentle" behavior that matches O2O dynamics, where costs are large and volatile at the beginning and gradually stabilize as the policy improves. All gains are clipped within shared bounds, and the integral channel includes anti-windup protection, ensuring stability. Although heuristic, this design follows standard gain-scheduling strategies that modulate proportional and integral aggressiveness according to operating conditions.
>
> **2. $K_d$**
> The derivative gain is updated using the short-horizon standard deviation of recent costs maintained in the same fixed-length `deque`:
> $$
> K_d \\leftarrow \\mathrm{clip}\\Big(K_d \\cdot (1 + \\gamma \\tfrac{\\sigma_c}{\\bar c}),\\, K_{d,\\min}, K_{d,\\max}\\Big),
> $$
> where $\\sigma_c$ is the standard deviation computed over a fixed-length deque (window size $d$). This rule makes $K_d$ non-decreasing by design, which is intentional and consistent with conservative gain scheduling for derivative damping. During the early finetuning phase, the cost signal is highly non-stationary and derivative noise is large; increasing $K_d$ enhances damping and prevents oscillations in $\\lambda_t$.
> As training proceeds and cost fluctuations diminish, the derivative term
>
> $$
> \partial_t = (J_c - J_{c,\\text{prev}})_+
> $$
>
> becomes small, so the effective derivative action weakens naturally even if $K_d$ remains high. The monotone non-decreasing design therefore ensures robustness in stochastic phases without causing over-damping later. All $K_d$ values stay within the bounded range $[K_{d,\\min}, K_{d,\\max}]$, and the windowed variance stabilizes as the system converges.  According to our observations, $K_d$ does not rapidly increase to $K_{d,\\max}$; it rises sharply during the early stage of online finetuning when the cost fluctuates significantly, and then increases only very slowly as the training stabilizes.

---

> ### Author Response · Authors · 2025-11-25
> **Response to Reviewer 5PFa**
>
> # **critical/main concerns on the PID control and its adaptation rules [part 3]: whether clipping bound in aPID same across all environment.**
>
> **A:** Yes, we confirm that the same clipping bounds are applied to aPID in all environments and they do important for training.
> Specifically, each PID gain is limited to the range from $0.1\times$ to $10\times$ its initial value, i.e.,
> $K_p \\in [0.1K_p^{(0)},\,10K_p^{(0)}]$,
> $K_i \\in [0.1K_i^{(0)},\,10K_i^{(0)}]$, and
> $K_d \\in [0.1K_d^{(0)},\,10K_d^{(0)}]$.
> These bounds were chosen to ensure numerical stability and are shared across all tasks, demonstrating that the method does not rely on environment-specific tuning.
>
> # **critical/main concerns on the PID control and its adaptation rules [part 4]: Clarification on $\\Delta t$ in PID approach**
>
> **A:** We set $\\Delta t = 1$, which corresponds to one online update step between successive PID updates. This discrete step size reflects the natural temporal resolution of the online training process, where each PID update occurs once per policy update iteration.
>
> # **Minor Concerns/Corrections on Theorem 1 [part 1]: Clarification on Confidence parameter $\delta$ and the high-probability claim**
>
> **A:** $\delta$ is the standard confidence parameter from a least-squares generalization bound (Lemma A.11 in [2]) used to control the one-step regression error. Applying a union bound over $k\\le K$ yields the factor $\log(|\\mathcal F|K/\delta)$ and the claim “with probability at least $1-\\delta$” simultaneously for all steps.
>
> # **Minor Concerns/Corrections on Theorem 1 [part 2]: Clarification on Choice of $\\alpha_r^{\\text{VPA}}$ and $\\alpha_c^{\\text{VPA}}$**
>
> **A:** The VPA target is
> $$
> y(\\alpha) = r + \\gamma\\mathbb{E}_{a'\\sim\\pi_0(\\cdot|s')}\big[Q(s',a') - \\alpha\\log\\pi_0(a'|s')\big].
> $$
> Larger $\\alpha$ increases the target on high-entropy (low-probability) actions under $\\pi_0$. For the **reward** critic, this yields an optimistic alignment that facilitates exploration at the beginning of online finetuning, which is desirable in the O2O setting. For the **cost** critic, the same mechanism inflates $Q_c$ on uncertain actions, which increases conservatism and can slow down finetuning when offline costs are sparse. In addition, the statistical term in Theorem 1 scales with $(1+\\alpha B)^2$, so a larger $\\alpha$ enlarges $\\tilde\\epsilon(\\delta,K)$. Combining these effects, we select $\\alpha_r^{\\text{VPA}}$ moderately larger to guide exploration and $\\alpha_c^{\\text{VPA}}$ smaller to keep the cost critic accurate and avoid excessive conservatism in subsequent Lagrangian updates. If $\\alpha_c^{\\text{VPA}}$ is large, the method becomes more conservative and convergence can be slower; our ablations indicate that a smaller $\\alpha_c^{\\text{VPA}}$ achieves a better accuracy–conservatism trade-off.
>
> # **Minor Concerns/Corrections on Theorem 1 [part 3]: Clarification on Approximation errors versus dataset size**
>
> **A:** In the statements around Theorem~1, the intended meaning is that the approximation errors **decrease as the dataset size increases**. In our bound this dependence appears through
>
> $$
> \tilde\epsilon(\delta,K)
> =\mathcal O\Big(\frac{\log(|\mathcal F|K/\delta)}{|\mathcal D|}\Big)+20\,d_F^{\pi_0,\mathrm{VPA}},
> $$
>
> so the statistical component decays at the usual $|\mathcal D|^{-1}$ rate inside the square root of the final error bound.  We will revise the text to state “decrease **as** the dataset size increases”.

---

> ### Author Response · Authors · 2025-11-25
> **Response to Reviewer 5PFa**
>
> # **Other Minor Concerns [part 1]: Relation between $L^{\\mathrm{CPQ}}_{Q_c}$ and Eqs. (6)–(7).**
>
> **A:** Eqs. (6)–(7) summarize the \emph{offline safe RL} critic-learning objective with an explicit OOD-support regularizer $P(\\cdot)$: the reward and cost critics $(Q,Q_c)$ are fit on the offline dataset while imposing pessimism on actions outside the data support. Concretely, in CPQ the term $P(\\cdot)$ is instantiated via a CVAE density model that down-weights/penalizes OOD actions, which yields pessimistic Bellman targets for such actions. Basically, Eqs. (6) and (7) represent the general objective of offline safe RL, while $L^{\\mathrm{CPQ}}_{Q_c}$ corresponds to the specific objective of a particular offline safe RL algorithm, CPQ.
>
> # **Other Minor Concerns [part 2]: Why can low predicted costs on in-dataset (IND) state–actions limit exploration?**
>
> **A:** Our statement concerns the **joint effect** of (i) sparse offline costs and (ii) offline pessimism for OOD actions. With sparse offline costs, the pretrained cost critic $Q_c$ tends to assign **low** costs on IND state–actions, while conservative offline learning (e.g., CPQ with CVAE) assigns **high**} costs to OOD state–actions via pessimistic extrapolation. Under a Lagrangian update, the policy then prefers to remain within the low-cost IND region and avoids regions that the critic deems high-cost (OOD), even when those regions contain high reward. Hence exploration beyond the offline support is **discouraged**. This is precisely where our O2O procedure helps: VPA aligns the critics to reduce value mismatch within the data support (optimistic for reward, conservative but accurate for cost), and the subsequent aPID-controlled finetuning safely increases exploration pressure while quickly damping violations.
> We will rephrase the sentence to make explicit that the limitation arises from **low IND predictions together with high OOD predictions**, not from low costs in isolation.
>
> # **Response to Other Minor Corrections**
>
> **A:** **Notation consistency for input distributions.** We have unified the notation across Secs. 2–3: $d$ denotes the online replay buffer distribution, while $D$ represents the distribution of $(s,a,s')$ in the offline dataset.
>
> **Right below Eq. (15), what is $c_t$?** We have clarified this in the revision to match the implementation.
> The variable $c_t$ now denotes the per-episode average cost at timestep $t$, which is appended to the fixed-length deque used by the PID controller. The deque is used to compute the mean $\\bar c$ and standard deviation $\\sigma_c$ of recent costs for adaptive gain updates. This aligns with the position-form PID description in the revised text:
>
> 'We use a discrete, position-form PID update with step size $\\Delta t=1$. Let the per-episode average cost be $c_t$ at timestep $t$, the limit be $c_{th}$, and the violation be $e_t = c_t - c_{th}$. Define an EMA-smoothed violation $\\tilde e_t = a_p \\tilde e_{t-1} + (1-a_p)e_t$ with smoothing factor $a_p\\in[0,1)$, and an EMA-smoothed cost $\\tilde c_t = a_d \\tilde c_{t-1} + (1-a_d)c_t$ with smoothing factor $a_d\\in[0,1)$, with a derivative buffer of length $d\\in\\mathbb N$. The integral state and derivative signal are $I_t=\\max(0,I_{t-1}+K_i e_t)$ and $D_t=\\max(0,\\tilde c_t-\\tilde c_{t-d})$, and the multiplier is $\\lambda_t=\\max(0,K_p\\tilde e_t+I_t+K_dD_t)$.'
>
> **$L_2$ notation in Eqs. (8) and (9).**
> We now add an explanation that $\\mathcal{L}_2$ means the standard Euclidean norm notation $\\lVert\\cdot\\rVert_2$.
>
> **Algorithm 1 (Appendix A): what is the final policy?** Algorithm 1 is written as a **generic, pluggable interface** for primal–dual online safe RL. This was intended to indicate that Marvel can be paired with any primal–dual online safe RL method (e.g., TRPO-lag, SAC-lag).
> In the revision we have (i) made the outputs explicit and (ii) instantiated the losses we actually use, **SAC-lag**, directly in the algorithm box.
>
> **pp.\,10 wording.** We corrected the typo “columns” $\\rightarrow$ “rows”.
>
> All above edits have been made in the revision; they are clarifications only and do not affect algorithms or results.

---

> ### Author Response · Authors · 2025-11-25
> **Response to Reviewer 5PFa**
>
> # **Response to Broader Impact Concerns**
>
> **A:** We appreciate the reviewer’s insightful comment on the safety aspect during the online learning phase. We fully agree that, as with most existing safe RL methods, absolute safety cannot be guaranteed during exploration, especially at early training stages. To better illustrate the trade-off between safety (measured by cumulative cost) and performance (maximum reward), we have included an additional analysis in Fig. 16. This figure shows that our proposed method, Marvel, achieves the highest maximum reward given a specific cumulative cost, and conversely, reaches a target performance level with the lowest cumulative cost. This demonstrates that although safety violations may still occur occasionally, Marvel significantly improves the reward–safety efficiency compared to baselines, thereby reducing the overall safety risk during training.
>
> [1] Stooke, Adam, Joshua Achiam, and Pieter Abbeel. "Responsive safety in reinforcement learning by pid lagrangian methods." International Conference on Machine Learning. PMLR, 2020.
>
> [2] Agarwal, Alekh, et al. "Reinforcement learning: Theory and algorithms." CS Dept., UW Seattle, Seattle, WA, USA, Tech. Rep 32 (2019): 96.
>
> [3] Liu, Zuxin, et al. "Datasets and benchmarks for offline safe reinforcement learning." arXiv preprint arXiv:2306.09303 (2023).

---

### Comment · Reviewer_5PFa · 2025-12-17
**Additional Concerns & Comments**

The authors properly addressed my comments and revised the paper accordingly, especially the PID part, although there are some additional comments that I summarised below that the authors need to consider to improve the paper.

1. On pp. 7, the step size $\Delta t = 1$ does not need to be there as it is never used in the explanation thereafter, i.e., it is just enough to say that “We use a discrete, position-form PID update”. When it is discretised, $\Delta t$ is integrated within the PID gains and usually $0 < \Delta t \ll 1$, not $\Delta t = 1$. So, *not* mentioning $\Delta t$ would be the best option.
2. It’d be good to explain, perhaps within Appendices, the additional design of the PID — “anti-windup (the maximum/clipping part? -- I'm not so sure about the connection between anti-windup and max/clip)”, “derivative lag”, “what does EMA stand for?”
3. $K_d$-scheduling (15) is understandable, but I don’t fully agree with it:
* a reasonable heuristic would be that when the std $\sigma_t$ becomes constant, $K_d$ doesn’t change;
* the learning environment can change in the middle and thereby becomes non-stationary again; then $K_d$ should be initialised again — it’d be good to mention this at some point, maybe the Limitations section in Appendix;
* $K_{d_\mathrm{min}}$ seems unnecessary as $K_d$ is non-decreasing.
4. In (8) and (9), the targets “$Q(s, a) - y$” and “$Q_c(s, a) - y$” are scalars with random variables $s$ and $a$ that are expected out by $\mathbb{E}_D[\cdot]$. It seems that L2 norm is not necessary.
5. Why are $\max_{a’} Q(s’, a’)$ and $\max_{a’} Q_c(s’, a’)$ in Eqns. (6) and (7)? Are they errata? Two policies maximising $Q$ and $Q_c$ are different; those bootstrapping terms, e.g., $\max_{a’} Q(s’, a’)$ make sense only in the unconstrained fashion. The corresponding terms in (8) and (9) with $a' \sim \pi_0( \cdot | s')$ are understandable, though.
6. On pp. 7, what does the *per-episode average* cost $c_t$ mean? Isn’t it just a cost $c_t$ obtained at time $t$ that is appended to the fixed-length deque used by the PID controller? If $c_t$ is already averaged (per-episode?) then, why are you averaging again to obtain $\tilde c_t$ and $\bar c_t$? Is the std $\sigma_c$ obtained not from the samples but from the (per-episode) averages?
7. In Algorithm 1, there is only one comment added. Still, I don’t know which object or policy Algorithm 1 returns as its outputs, nor the notations. Are the notations all employed from CPQ (Xu et al., 2022) and SAC-lag (Ray et al., 2019)? Even if so, the authors can make them more explicit and clearer as Algorithm 1 is located in Appendices that have almost no page limit.
8. Adding Fig. 16 is great. I appreciate the effort to improve the broader impact.

---

> ### Author Response · Authors · 2025-12-18
> **Response to Additional Concerns and Comments**
>
> We sincerely thank Reviewer 5PFa for acknowledging our revisions and for providing additional constructive comments that further strengthen the paper.
>
> 1. We thank the reviewer for the suggestion. We have polished the presentation accordingly in the revision.
>
> 2. We have added a short clarification in Appendix F explaining the additional PID design components, including anti-windup via clipping and derivative smoothing via EMA. These are standard choices in discrete PID implementations and help improve numerical stability. The new text is highlighted in the revision.
>
> 3. We appreciate the reviewer’s thoughtful comments on the $K_d$-scheduling mechanism.
>
> (a) Regarding the observation that ``when $\\sigma_t$ becomes constant, $K_d$ should stop changing,'' we agree. In practice, once the cost dynamics stabilize, $\\sigma_t$ becomes small and the multiplicative update $(1+\\gamma\\sigma_t/\\bar c)$ stays extremely close to 1. As a result, $K_d$ naturally plateaus, and we did not observe drift after stabilization. We have added a remark noting this behavior.
>
> (b) The reviewer is also correct that re-initializing $K_d$ may be helpful if the environment becomes non-stationary again. In our experiments, non-stationarity primarily arises during the early online adaptation phase, where $K_d$ adjusts rapidly through volatility-based updates. After convergence, renewed non-stationarity was not observed(in our experiments). For completeness, we now mention in the appendix I that re-initializing $K_d$ is a possible extension when strong non-stationarity is expected.
>
> (c) We agree that $K_{d,\\min}$ is rarely active because $K_d$ is monotonically non-decreasing by design. It is included only for consistency with $(K_{p,\\min}, K_{i,\\min})$. We will clarify this point in the revision.
>
> 4. We appreciate the reviewer’s suggestion. Since $Q(s,a)-y$ and $Q_c(s,a)-y_c$ are scalar quantities, the $\\ell_2$ notation is unnecessary. We have replaced it with the standard squared-error form for clarity.
>
> 5. Thank you for pointing this out. The reviewer is correct that the Bellman backups in offline safe RL should be expectation-based. The $\\max_{a'}$ terms in Eqs. (6)-(7) were a typographical mistake inherited from an early draft. In the revision, we corrected them to the intended expectation forms in revision.
>
> 6. We have clarified the definition of $c_t$. Here $t$ indexes training iterations, each corresponding to a full episode. The value $c_t$ is the total episode cost, consistent with the safe-RL constraint, which is defined on the expected episode cost. The aPID controller therefore operates on EMA-smoothed episode-level costs. A detailed explanation is added in the revision.
>
> 7. We thank the reviewer for this helpful feedback. Algorithm 1 has been substantially revised to (i) explicitly specify all inputs, including offline/online loss functions, separate learning rates, and the initial aPID controller state, and (ii) clearly list all outputs. The updated algorithm further clarifies that Marvel is a modular framework: the offline and online losses define a generic actor-critic safe RL method, and CPQ/SAC-lag are only instantiations used in our experiments. The revised Algorithm 1 is much clearer and we believe it fully resolves the reviewer’s concern.
>
> 8. We thank the reviewer for the positive feedback.
>
> We sincerely thank the reviewer again for the constructive and insightful feedback. The suggested clarifications have helped us significantly improve the presentation and rigor of the paper. We hope the revised version addresses all concerns and meets the reviewer's expectations.

---

> > ### Comment · Reviewer_5PFa · 2025-12-27
> > **Reply to the response to Additional Concerns and Comments.**
> >
> > The reviewer is satisfied with the response and will submit official recommendations soon. To further improve the paper, consider my minor suggestions below.
> >
> > 6. Now, I understand better. But, there is a confusion between the discounted cost $\sum_{t=0}^\infty \gamma^t c(s_t, a_t)$, which defines $C(\pi)$ in Sec. 2, and the total episodic cost $c_t$. The confusion is due to the word 'total', which sometimes means "undiscounted". I would suggest to add "discounted" to the sentence and/or give a precise formula $C_t = \sum_{t=0}^\infty \gamma^t c(s_t, a_t)$, where I'd also suggest to write $C_t$ instead of $c_t$ for a match with $C(\pi)$ (which however may cause changes in other places and notations like changing $\tilde c_t$ and $\bar c_t$ to $\tilde C_t$ and $\bar C_t$ --- so it's up to the authors).
> >
> > On pp. 7, I believe you don’t need to say “the controller does not introduce a second integrator” anymore as the issue has been resolved by the revision.

---

> > > ### Author Response · Authors · 2025-12-27
> > > **Reply to Reviewer 5PFa**
> > >
> > > We thank the reviewer for the careful reading and constructive suggestions. As suggested, we have clarified in the paper that the cost is discounted to avoid any potential ambiguity. In addition, we have removed the sentence regarding the second integrator, as it is no longer necessary after the revision.

---

### Decision · Action_Editor_zVRU · 2026-01-04

**Recommendation:** Accept with minor revision

**Additional Comments:**

I suggest the authors prepare the final proofs for the concerns, including the discussions with Reviewer 5PFa. Subject to checking the proofs, the paper can be accepted.

**Audience:**

Yes

**Audience Explanation:**

It is of interest to some of the TMLR's audience who aims to understand how to leverage offline data to improve sample efficiency of online learning.

**Claims And Evidence:**

Yes

**Claims Explanation:**

The claims are generally presented in a way that met the reviewers' expectations,  and the reviewers mentioned that the authors addressed the concerns. However, the main remaining question is that the amount of changes means that the authors need to thoroughly check again the proofs and then submit the camera-ready copy. Then, the AE will be able to confirm that the comments are duly addressed.